# Antiprotozoal and Antitumor Activity of Natural Polycyclic Endoperoxides: Origin, Structures and Biological Activity

**DOI:** 10.3390/molecules26030686

**Published:** 2021-01-28

**Authors:** Valery M. Dembitsky, Ekaterina Ermolenko, Nick Savidov, Tatyana A. Gloriozova, Vladimir V. Poroikov

**Affiliations:** 1Centre for Applied Research, Innovation and Entrepreneurship, Lethbridge College, 3000 College Drive South, Lethbridge, AB T1K 1L6, Canada; nick.savidov@lethbridgecollege.ca; 2A.V. Zhirmunsky National Scientific Center of Marine Biology, 17 Palchevsky Str., 690041 Vladivostok, Russia; ecrire_711@mail.ru; 3Institute of Biomedical Chemistry, 10 Pogodinskaya Str., 119121 Moscow, Russia; tatyana.gloriozova@ibmc.msk.ru (T.A.G.); vladimir.poroikov@ibmc.msk.ru (V.V.P.)

**Keywords:** antiprotozoal, antitumor, polycyclic, peroxides, pharmacological potential, PASS

## Abstract

Polycyclic endoperoxides are rare natural metabolites found and isolated in plants, fungi, and marine invertebrates. The purpose of this review is a comparative analysis of the pharmacological potential of these natural products. According to PASS (Prediction of Activity Spectra for Substances) estimates, they are more likely to exhibit antiprotozoal and antitumor properties. Some of them are now widely used in clinical medicine. All polycyclic endoperoxides presented in this article demonstrate antiprotozoal activity and can be divided into three groups. The third group includes endoperoxides, which show weak antiprotozoal activity with a reliability of up to 70%, and this group includes only 1.1% of metabolites. The second group includes the largest number of endoperoxides, which are 65% and show average antiprotozoal activity with a confidence level of 70 to 90%. Lastly, the third group includes endoperoxides, which are 33.9% and show strong antiprotozoal activity with a confidence level of 90 to 99.6%. Interestingly, artemisinin and its analogs show strong antiprotozoal activity with 79 to 99.6% confidence against obligate intracellular parasites which belong to the genera Plasmodium, Toxoplasma, Leishmania, and Coccidia. In addition to antiprotozoal activities, polycyclic endoperoxides show antitumor activity in the proportion: 4.6% show weak activity with a reliability of up to 70%, 65.6% show an average activity with a reliability of 70 to 90%, and 29.8% show strong activity with a reliability of 90 to 98.3%. It should also be noted that some polycyclic endoperoxides, in addition to antiprotozoal and antitumor properties, show other strong activities with a confidence level of 90 to 97%. These include antifungal activity against the genera Aspergillus, Candida, and Cryptococcus, as well as anti-inflammatory activity. This review provides insights on further utilization of polycyclic endoperoxides by medicinal chemists, pharmacologists, and the pharmaceutical industry.

## 1. Introduction

Polycyclic endoperoxides are a rare group of naturally occurring metabolites found in various parts of plants such as leaves, roots, bark, stems, seeds, fruits, and flowers [1,2,3,4,5,6,7,8,9,10,11,12,13,14,15,16,17]. In addition, they have been found in extracts of various types of marine invertebrates and algae, and endoperoxides are synthesized by various types of fungi, fungal endophytes, and other microorganisms [8,9,13,14,15,18,19,20].

As shown in recent years, many polycyclic endoperoxides, both natural and synthetic, have antimalarial effects [21,22]. It is known that malaria or “swamp fever” refers to a group of transmissible infectious diseases transmitted to humans by bites of female mosquitoes belonging to the genus Anopheles, caused by parasitic protists of the genus Plasmodium, mainly *P. falciparum* [23,24,25]. According to the WHO World Malaria Report, at the beginning of the 21st century, the incidence ranged from 350 to 500 million cases per year, of which 1 to 3 million ended in death [26,27]. In connection with these ominous data, any new sources of natural antimalarial agents are of great interest to medicine and pharmacology, as well as to the pharmaceutical industry [28,29,30].

In this review, we will look at rare and unusual polycyclic endoperoxides isolated from different terrestrial and marine sources. The biological activity of many polycyclic endoperoxides has not been determined, and we present the pharmacological activities detected experimentally and predicted based on the structure-activity relationships using the PASS (Prediction of Activity Spectra for Substances) software [31,32,33]. PASS estimates the probabilities of several thousand biological activities with an average accuracy of about 96%. Probability of belonging to the class of “actives” Pa is calculated for each activity, providing the assessment of the hidden pharmacological potential of the investigated natural polycyclic endoperoxides [2,7,13,14,17,31,32,33].

## 2. Polycyclic Endoperoxides Derived from Marine Sources

Marine algae (both microalgae and macrophytes) and invertebrates are the main source of biologically active secondary metabolites, which include hydrocarbons, terpenoids, lipids, steroids, carotenoids, aromatic compounds, and alkaloids, as well as mixed compounds containing heteroatoms and polycyclic endoperoxides [2,4,5,6,7,8,17,18,19,20,34,35,36,37,38,39,40,41,42,43,44,45,46,47,48,49,50,51,52,53,54,55,56,57,58,59,60,61,62,63,64].

A series of polycyclic peroxides such as contrunculin B (**1**) as well as the trunuclin peroxides (**2**–**7**) were discovered in the extracts of Australian marine sponge *Latrunculia conulosa* [65], *Latrunculia* sp. [66] and found in an Okinawan sponge *Sigmosceptrella* sp. [67]. Structures (**1**–**16**) can be seen in Figure 1, and their biological activity is presented in Table 1. Two unusual endoperoxide diterpenoids (**8** and **9**) were isolated from the brown seaweed *Taonia atomaria* [68]. Cytotoxic 8,11-epidioxy-7-hydroxy-3,12,15(17)-cembratrien-16,2-olide called cembranolide C (or denticulatolide, **10**) known as icthyotoxin was found in soft corals *Lobophytum denticulatum, Sinularia mayi*, and *Sarcophyton crassocaule* and its acetate (**11**) was also found in *L. denticulatum* extract [69,70,71,72].

Norditerpenoid, aplypallidioxone (**12**) was detected in Australian encrusting sponge *Aplysilla pallida* [73], and two abietic acids (**13** and **14**) that were previously found in plants have also been found in green algae *Elodea canadensis* [74].

A guaiane-type sesquiterpene, 1,7-epidioxy-5-guaiene (**15**) was found and later isolated from *Axinyssa* sponge [75], and an oxygenated sesquiterpenoid, 1,7-epidioxy-5-guaien-4-ol called peroxygibberol (**16**), was isolated from a Formosan soft coral, *Sinularia gibberosa*, which demonstrated moderate cytotoxicity toward a human liver carcinoma cell line [76]. Structures (**16–35**) can be seen in Figure 2, and their biological activity is presented in Table 2.

An extract of a marine sponge, *Lendenfeldia chondrodes* has led to the isolation and identification of two C-24 stereoisomers (**17** and **18**) of steroid, 5*R*,8*R*-epidioxy-24-hydroperoxy-cholesta-6,28(29)-dien-3α-ol. Obtained data with the molecular formula of steroid indicated that a hydroperoxy group and a vinyl group are attached at position-24 in both the *R*- and *S*- configurations [77], and cytotoxic steroid, (3β,5α,8α,24*R*,25*R*)-epidioxy-24,26-cyclocholesta-6,9(11)-dien-3-ol (**19**) was identified from *Tethya* sp. [78].

Interestingly, steroid, (3β,5α,8α)-epidioxycholest-6-en-3-ol (**20**) was found in three cone snail species, *Conus ebraeus, C. leopardus,* and *C. tessulatus* (family Conidae) [79], and was also present in the extract of polychaete worm *Perinersis aibuhitensis* [80], it was also isolated from the steroid fraction of sponges *Axinella cannabina*, *Luffariella* cf. *variabilis* [81,82], the tunicate *Cynthia savignyi* [83], and in long-spined sea urchin *Diadema setosum* [84]. Isolated steroid showed antibacterial, antifungal, and cytotoxic activities [81,82,83,84]. Detection of this steroid in various species of marine invertebrates could indicate that they all share a food chain, and the source of this steroid may be algae.

(3β,5α,8α)-Epidioxy-24-methylenecholest-6-en-3-ol (**21**) has been isolated from the several marine invertebrates, tunicate *Ascidia nigra*, pillar coral *Dendrogyra cylindrus*, marine sponge *Thalysias juniperina*, and sea hare *Aplysia dactylomela* [85]; in addition, this steroid was found in the tunicates *Dendrodoa grossularia* and *Ascidiella aspersa*, the gastropoda *Aplysia depilans* and *Aplysia punctata* [86], the sea anenome *Metridium senile* [87], and the sponge *Tethya aurantia* [88]. 

(3β,5α,8α,*22E,24S*)-Epidioxy-24-methylcholesta-6,22,25-trien-3-ol called axinysterol (**22**), (3β,5α,8α,24*R*)-Epidioxy-24-methylcholest-6-en-3-ol (**23**) and (3β,5α,8α,22*E*,24*R*)-Epidioxystigmasta-6,22-dien-3-ol (**24**) were detected in MeOH extract of the marine sponge *Luffariella* cf. *variabilis* [85].

22,23-Dihydro-5,8-epidioxystigmast-6-en-3-ol (**25**) was surrounded by *Luffariella* cf. *variabilis, Tethya* sp., and sea squirt *Dendrodoa grossularia* [82,85,86,87,88]. (3β,5α,8α)-Epidioxy-22,23-cyclopropacholest-6-en-3-ol (**26**) and (3β,5α,8α)-endoperoxy-23-demethylgorgost-6-en-3-ol (**27**) were discovered in soft corals *Sinularia maxima*,* S. gibberosa* and *Sinularia* sp. [89,90].

(3β,5α,8α,22*E*,24*S*)-Epidioxyergosta-6,9(11),22-trien-3-ol (**28**) was found in two tunicates *Ascidia nigra* and *Dendrogyra cylindrus* and sponge *Thalysias juniperina* [82,85,91], and (3β,5α,8α)-epidioxy-24-methylcholesta-6,9(11),24(28)-trien-3-ol (**29**) was detected in *Ascidia nigra* [85,88].

(3β,5α,8α,22*E*,24*R*)-Epidioxy-23,24-dimethylcholesta-6,22-dien-3-ol (**30**) was isolated from MeOH extract of the single-celled algae *Odontella aurita* [92], and it was also found in edible mushrooms *Lentinus edodes*, which are also known as shiitake [93].

(3β,5α,8α,22*E*)-Epidioxy-24-norcholesta-6,22-dien-3-ol (**31**) was detected in the sea pen, opisthobranch mollusk *Virgularia* sp. [94], and in *A. nigra, D. cylindrus*, and *T. juniperina* [85]. (3β,5α,8α,24(28E))-Epidioxy-24-ethylcholesta-6,24(28)-dien-3-ol (**32**) has been isolated and structure elucidated from several tunicates, namely *Ascidia nigra* and *Dendrogyra cylindrus*, and (3β,5α,8α,24(28)*Z*)-form was detected in *Dendrodoa grossularia* [85,86].

Cytotoxic (3β,5α,8α,22*E*,24*R*)-epidioxyergosta-6,22-dien-3-ol (**33**), well-known as 5α,8α-peroxyergosterol, is the most widely distributed steroid in the plant kingdom, lichens and fungi [5,6], and is also found in marine sponges *Axinella cannabina*, *Halichondria* sp., *Suberites carnosus*, *Spirastrella abata*,* Thalysias juniperina* [85,95,96,97], the sea lily *Gymnocrinus richeri* [98], and tunicates *Ascidia nigra*, *Dendrogyra cylindrus* [88].

Two cytotoxic steroids, 5α,8α-epidioxy-cholesta-6,9(11),24-trien-3β-ol (**34**) and 5α,8α-epidioxy-cholesta-6,23-dien-3β,25-diol (**35**) were isolated from a marine sponge *Monanchora* sp. [99]. Series 5α,8α-epidioxysteroids: **20**, **21**, **23**, **32**, and **36**–**41** were isolated from the MeOH extracts of the Gorgonian *Eunicella cavolini* and the tunicate *Trididemnum inarmatum*. Compound (**36**), bearing a cyclopropyl moiety in the side chain, exhibited the highest antiproliferative activity [100]. Structures (**36**–**41**) can be seen in Figure 3, and their biological activity are presented in Table 3.

## 3. Polycyclic Endoperoxides Derived from Fungi and Fungal Endophytes

Fungi, fungal endophytes, myxomycetes, and the lichenized Ascomycetes are of great interest to pharmacologists and chemists, since they produce many biologically active substances, such as aromatic and phenolic compounds, tannins, hydrocarbons, lipids, unusual steroids, triterpenoids, heterocyclic compounds, peptides, and polycyclic endoperoxides [101,102,103,104,105,106,107,108,109,110,111,112,113,114].

In fungi, both cultivated and wild, polycyclic endoperoxides are found in small quantities, but ergosterol peroxide (**33**) is the most abundant [5,6]. Below, we present data on the distribution of this steroid and other polycyclic endoperoxides in fungi, fungal endophytes and lichens.

Trung and co-workers [115], using a modernized quantitative high-performance liquid chromatography method, found that ergosterol peroxide is present in wild mushrooms such as *Fomitopsis dochmius*,* F. carneus*,* Daldinia concentrica*,* Ganoderma applanatum*,* G. lobatum*,* G. multiplicder G. lucidum*, *Phellinus igniarius*, and *Trametes gibbosa*. In addition, this steroid has been detected in other species of wild fungi, fungal endophytes and lichens: *Claviceps purpurea, Ganoderma lucidum*,* G. tsugae*,* G. sichuanense*, *Daedalea quercina*, *Piptoporus betulinus*,* Cryptoporus volvatus*, *Guignardia laricina, Lampteromyces japonicus*, *Botrytis cinerea*, *Lactarius uvidus*, *L. volemus*,* Cryptoporus volvatus*,* Dictyonema glabratum*,* Lasiosphaera nipponica*,* Gloeophyllum odoratum*,* Gymnopilus spectabilis*,* Hericium erinaceus*,* Hypsizigus marmoreus*,* Inonotus obliquus*,* I. radiatus, Lenzites betulina*,* Meripilus giganteus*,* Microporus flabelliformis*,* Naematoloma fasciculare*, *Phellinus pini*,* P. ribis*,* P. torulosus*,* Roseoformes subflexibilis*,* Pyropolyporus fomentarius*,* Pisolithus tinctorius*,* Polyporus tuberaster*, *Pseudephebe pubescens* [5,6,17,116], and from the edible mushroom *Volvariella volvacea* [117]. In addition, ergosterol peroxide has been found in some Ascomycetes, *Aspergillus* sp., *A. niger*, *A. oryzae*, *A. flavus*, *A. terreus*, and *A. fumigatus*, *Fusurium monilforme*, *F. osysporum*, *Penicillium rubrum*, and *P. sclerotigenum* [5]. Ragasa [118] researched Philippine mushrooms and found ergosterol peroxide in *Auricularia auricula-judae*, *Coprinopsis lagopus*, *Pleurotus florida*, and *Phellinus gilvus*.

It is known that ergosterol peroxide isolated from edible or medicinal mushrooms demonstrates antitumor activity against colorectal cancer, hepatocellular carcinoma, prostate cancer, myeloma, and leukemia [119,120,121,122,123], and it also possesses antioxidant, anti-inflammatory, and antiviral activities, as well as induce the apoptosis of cancer cells [124,125,126,127,128].

Endoperoxide (**42**), bearing a keto group at the 12 position, has been isolated from the fungus *Fusarium monilforme* [129]. Endoperoxy glycoside (**43**) was detected in ethanol extract of the fungus *Lactarius volemus*, which demonstrated anticancer activity [130,131]. Ergosterol peroxide (**33**) and unusual steroid called asperversin A (**44**) have been isolated from endophytic fungus of *Aspergillus versicolor* that was isolated from the seaweed *Sargassum thunbergii*. Both steroid antibiotics showed antibacterial activity against *Escherichia coli* and *Staphylococcus aureus* [132], and another steroid named fuscoporianol D (**45**) was found in a MeOH extract of in field-grown mycelia of *Inonotus obliquus* [133].

Several steroids containing a 5,9-position peroxide moiety have been isolated from some mushroom extracts. For example, endoperoxide (**46**) was found in *Boletus calopus* white mushroom [134], and steroid (**47**) produces by two fungi *Panellus serotinus* and *Lepista nuda* [135]. Two steroids named nigerasterols A and B (**48** and **49**) were isolated from the extracts of an endophytic fungus of *Aspergillus niger* MA-132, which was isolated from the mangrove plant *Avicennia marina* [136], and steroids (**49**–**52**) were found in *Buna shimeji* and *Pleurotus ostreatus* [137]. A rare chamigrane-type sesquiterpenes called steperoxides A (**53**), B (**54**), C (**55**), and D (**56**) have been isolated from the hydnoid fungus *Steccherinum ochraceum* [Phanerochaetaceae]. Compound (**53**) demonstrated anticancer properties, and compounds (**54** and **57**) showed significant antimicrobial activity against *Staphylococcus aureus* [138,139,140,141]. Structures (**53**–**68**) can be seen in Figure 3, and their biological activity are presented in Table 4.

Anti-tumor nor-sesquiterpene endoperoxides called talaperoxide A (**57**), B (**58**), C (**59**), and D (**60**) were isolated from culture of fungi *Talaromyces* species HN21-3C, and from a mangrove endophytic fungus, *Talaromyces flavus* isolated from the mangrove plant *Sonneratia apetala* [142]. Isolated fungal metabolites demonstrated antineoplastic activity against MCF-7, MDA-MB-435, HepG2, HeLa, and PC-3 cancer cell lines [143,144]. Semi-synthetic derivative (**61**) of the fungal derived natural product showed the antiparasitic and cytotoxic activity against *Trypanosoma brucei* and Hela cells, respectively [145]. Chamigrane endoperoxide named merulin C (**62**), were isolated from the culture broth extract of an endophytic fungus of *Xylocarpus granatum* [146].

Caryophyllene-derived meroterpenoids, called cytosporolides A (**63**), B (**64**), and C (**65**), which have a unique peroxylactone skeleton, were isolated from cultures of the fungus *Cytospora* sp. Obtained metabolites demonstrated significant antimicrobial activity against the Gram-positive bacteria *Staphylococcus aureus* and *S. pneumonia* [147].

Two unprecedented spiroketal endoperoxides named chloropupukeanolides A (**66**) and B (**67**) were isolated from an endophytic fungus *Pestalotiopsis fici*. Compound (**66**) showed significant anti-HIV-1 and cytotoxic effects [148].

It is known that natural hypocrellin is a dark red dye with photodynamic activity against several microorganisms was isolated from the fungus *Hypocrella bambusae*, and its photooxidation product called peroxyhypocrellin (**68**) has an anthracene endoperoxide arrangement within the perylene quinone structure [149]. Structures (**42**–**68**) can be seen in Figure 3 and Figure 4, and their biological activity is presented in Table 3 and Table 4.

## 4. Polycyclic Endoperoxides Derived from Plants and Liverworts

The largest amount of endoperoxides has been found, isolated and partially biological activity determined in plants and liverworts [1,2,5,6,8,9,14,19,150,151,152,153,154].

A peroxide-sesquetepene, called nardosaldehyde (**69**) was isolated from the roots of *Nardostachys chinensis*, and biological activity was not determined [155]. Structures (**69–89**) can be seen in Figure 5, and their biological activity is presented in Table 5 and Table 6. Peroxygibberol (**16**) is marine peroxide (5.9%) was also found in Agarwood oil obtained from highly infected *Aquilaria malaccensis* wood [156].

An antimalarial guaiane-type sesquiterpenoids (**70**), nardoperoxide (**71**), and isonardoperoxide (**72**) were isolated from *Nardostachys chinensis* roots [157,158,159], and in addition to this, nardoguaianone A (**73**), B (**74**), C (**75**), and D (**76**) were also highlighted from the same plant [160].

Widdarol peroxide (**77**) and its analogue (**78**) were found in hexane extract from the fruits of *Schisandra grandiflora*, which showed anti-proliferative activity against Hela (cervical cancer), A549 (lung cancer), DU-145 (prostate cancer), and MCF-7 (breast cancer) cancer cell lines [161].

Polycyclic sesquetepene, 1α,8α-epidioxy-4α-hydroxy-5αH-guai-7(11),9-dien-12,8-olide (**79**), which has anti-influenza viral properties, were isolated from the plant *Curcuma wenyujin*, which is mainly in the Wenzhou region of China [162] and was recently found in the flowering plant *Acorus calamus* [163]. Diterpenoid, (*E*,*E*)-15-hydroxylabda-8(17),11,13-trien-16-al (**80**) was detected in an extract of *Alpinia chinensis* [164]. Cadinane sesquiterpene, (−)-(5*S*,6*S*,7*S*,9*R*,10*S*)-7-hydroxy-5,7-epidioxycadinan-3-ene-2-one (**81**) was isolated and identified from the aerial part of the invasive plant *Eupatorium adenophorum* [165]. Diterpenoids called mulinic acid (**82**) and 17-acetoxymulinic acid (**83**) have been isolated from the aerial parts of *Mulinum crassifolium* (Umbelliferae) [166,167], and semi-synthetic derivatives (**94**, **95** and **96**) were obtained from mulinic acid [168,169].

A cytotoxic seven-membered endoperoxide hemiacetal called coronarin B (**84**) was isolated from the flowers of *Alpinia chinensis* and *Hedychium coronarium* [164,170,171]. Unusual diterpene peroxide (**85**), with potent activity against *Plasmodium falciparum*, has been isolated from *Amomum krervanh* [172].

Endoperoxide called artemisinin (**86**) was found in 1979 in the extract of the Chinese herb qinghaosu (*Artemisia annua*) [173]. Currently, artemisinin and its derivatives (**87**–**93**) are widely used throughout the world as antimalarial drugs against the protozoan parasites [174,175,176,177]. An interesting mechanism of action for these compounds appears to involve heme-mediated degradation of the endoperoxide bridge to form carbon-centered free radicals, and these free radicals are selectively toxic to malaria parasites [178,179,180]. Artemisinin and its derivatives exhibit antitumor, antifungal, and other activities [181,182,183]. Structures (**90**–**114**) can be seen in Figure 6, and their biological activity is presented in Table 6 and Table 7.

Endoperoxy cuparene-type sesquiterpenoids (**97** and **98**, structures are shown in Figure 6, and activity is shown in Table 7) were identified from the Japanese liverwort *Jungermannia infusca* [184,185]. The chamigranes called merulin B (**99**) and C (**100**) have been found in an extract of the culture broth of a Thai mangrove-derived fungus [186,187].

Muurolane sesquiterpene endoperoxide, 1,4-peroxy-5-hydroxy-muurol-6-ene (**101**) has been obtained from plant *Illicium tsangii* (family Schisandraceae) [188,189,190]. The peroxide called schisansphene A (**102**) was isolated from the plant *Schisandra sphenanthera*, also known as the magnolia berry [191].

Highly oxygenated sesquiterpene (+)-muurolan-4,7-peroxide (**103**) was found in the essential oil of the liverwort *Plagiochila asplenioides* [192], and two sesquiterpene endoperoxides (**104** and **105**) were isolated from the aerial parts of the invasive plant *Eupatorium adenophorum* [193,194]. Unusual endoperoxide (**106**) was detected in the *Ligularia veitchiana* [195], compound (**107**) was isolated from the leaves of *Eupatorium adenophorum* [196], and metabolite (**108**) was found in extracts of the *Xylopia emarginata* [197]. The aerial parts of *Montanoa hibiscifolia* afforded rare endoperoxide (**109**) [198].

The xanthane-type sesquiterpenoid 4β,5β-epoxyxanthatin-1α,4α-endoperoxide (**110**) was found in the aerial parts of *Xanthium strumarium* [199], and 2α,5α-endoperoxide (**111**), which possess the 6α,12-eudesmanolide structure, was detected in areal parts of the *Artemisia herba-alba* [200]. The sesquiterpene peroxide (**112**) has been found from the aerial parts of *Croton arboreous* [201]. 

Allohimachalane peroxide (**113**) has been obtained from *Illicium tsangii* [188,189,190], and an unusual sesquiterpene lactone with endoperoxide group, called tehranolide (**114**) with strong antimalarial activity has been discovered in many Iranian Artemisia species: *A. aucheri*, *A. austriaca*, *A. biennis*, *A. campestris*, *A. deserti*, *A. diffusa*, *A. gypsacea*, *A. haussknechtii*, *A. kermanensis*, *A. kopetdaghensis*, *A. kulbadica*, *A. oliveriana*, *A. persica*, *A. santolina*, *A. sieberi*, *A. tschernieviana*, *A. ciniformis*, *A. incana*, *A. turanica*, and *A. tournefortiana* [202].

The hemiacetal of tricycloperoxyhumulone A (**115**) was detected in hops (*Humulus lupulus*) [203]. Structures (**115**–**128**) can be seen in Figure 7, and their biological activity is presented in Table 8. Highly oxygenated limonoid featuring an unprecedented 3,4-peroxide-bridged A-seco skeleton called walsuronoid A (**116**) was isolated from *Walsura robusta* (family Meliaceae). The isolated peroxide showed weak antimalarial activity [204].

A cytotoxic peroxytriterpene dilactone called pseudolarolide I (**117**) has been isolated from the seeds of *Pseudolarix kaempferi* [205], and the leaves of *P. kaemferi* contains three triterpene peroxides, pseudolarolides Q (**118**), R (**119**), and S (**120**) [206]. An unusual glycoside, 3β,15α,25-trihydroxy-16,23-dioxo-6α,19α-epidioxy-9,10-seco-9,19-cyclolanost-5 (10),9(11)-diene 3-O-α-1-arabinopyranoside called podocarpaside E (**121**), was isolated from the roots of *Actaea podocarpa* [207].

A triterpene, 5α,6α-epidioxy-5β,6β-epoxy-9,13-dimethyl-25,26-dinoroleanan-3β-ol acetate, called aceranol acetate (**122**), which shows anti-inflammatory activity, was isolated from the stems and leaves of *Acer mandshuricum* [208]. The isolated compound also exhibited moderate activity against four human cancer cell lines (HL-60, SK-OV-3, A549, and HT-29).

A peroxy-multiflorane triterpene ester, (3α,5α,8α,20α)-5,8-epidioxymultiflora-6,9(11)-diene-3,29-diol 3,29-dibenzoate (**123**), was isolated from the processed seeds of *Trichosanthes kirilowii*. The obtained compound showed in vitro cytotoxicity against human-tumor cell lines (Hela, HL-60, and MCF-7) [209]. A peroxy triterpene, 3β-acetoxy-1β,11α-epidioxy-12-ursene (**124**), was isolated from the aerial roots of *Ficus microcarpa* [210]. An antimicrobial triterpenoid, 1α,5α-dioxy-11α-hydroxyurs-12-en-3-one (**125**), was found and obtained from the rhizome of *Vladimiria muliensis* [211]. 

The benzene extract of the bark of *Sapium baccatum* contained the nor-triterpene peroxide baccatin (**126**), which has been isolated and studied [212]. Two highly oxygenated ursane-type triterpenoids, (2β,3β)-3,25-epidioxy-2,24-dihydroxyursa-12,20(30)-dien-28-oic acid (**127**) and (2β,3β)-3,25-epidioxy-2,24-dihydroxyurs-12-en-28-oic acid (**128**), were detected in the EtOH extract of *Gentiana aristata* [213].

Highly oxygenated steroidal metabolites called physalin K (**129**) and Q (**130**) were found in extracts of the areal parts of *Physalis alkekengi* var. *franchetii* [214]. Structures (**129**–**143**) can be seen in Figure 8, and their biological activity is presented in Table 9. Plant withanolide called jaborosalactone 15 (**131**) was isolated from the flowering plant *Jaborosa odonelliana*, which was collected during autumn in Argentina [215]. Physangulidine G (**132**) was isolated from the aerial parts of *Deprea bitteriana*, *D. cuyacensis*, and *D. zamorae* [216].

A unique compound, a 3,9-(1,2,3-trioxocine)-type steroid called rauianodoxy (**133**), and an ergosterol peroxide (**33**) were isolated from the Australian plant *Rauia nodosa* (family Rutaceae) [217]. An unusual endoperoxide called schinalactone A (**134**), which has a compressed ring A and shows anticancer activity against PANC-1 cell lines, was detected in the stems and roots of the magnolia vine, *Schisandra sphenanthera* [218].

A secoadianane-type steroid (**135**) was found and identified in the herbaceous plant *Dorstenia brasiliensis* (Moraceae) [219,220]. A polycyclic peroxide called vielanin D (**136**), which showed anti-plasmodial activity, was extracted from fresh and dry leaves of the plant *Senecio selloi* [221]. The peroxy steroid (16*S*,23*R*)-16,23-epoxy-23,25-epidioxycycloartan-3-one (**137**) was found in the Texas yellow-star, *Lindheimera texana* (Asteraceae) [222].

Two triterpenes, called gilvanol (**138**) and 3-deoxydilvanol (**139**), have been detected in the extracts of the red-bark oak, *Quercus gilva* [170,223]. An interesting endoperoxide, adian-5-ene ozonide (**140**) was found in the fern leaves of *Adiantum monochlamys* (Pteridaceae) and *Oleandra wallichii* (Davalliaceae), and another peroxide, a triterpene ozonide (**141**), was detected in the root extract of *Senecio selloi* [224,225].

Interesting and rare 1,2,4-trioxolanes (**142** and **143**) were derived from natural two allobetulin derivatives; however, biological activity has not been determined [226]. Two 9,13-diepoxy labdane diterpenoids called amoenolide K (**144**) and its 19-acetate (**145**) were detected in the areal parts from *Amphiachyris amoena* [227], and ent-8β,12α-epidioxy-12β-hydroxylabda-9(11),13-dien-15-oic acid γ-lactone (**146**) was obtained from the aerial parts of *Premna oligotricha* [228]. Structures (**144**–**154**) can be seen in Figure 9, and their biological activity is presented in Table 10.

Diterpenic acids (**13**, **14**, **147** and **148**) have been identified from lipid extract of the different species. The diterpenic acid methyl ester (**147**) was isolated from the leaves of Moroccan *Juniperus thurifera* and *J. phoenicea* [229], compound (**148**) was detected in MeOH extract of *Safvia oxyodon* [230], and two abietic acids (**13** and **14**) were obtained from areal parts from the *Abies marocana*, *Lepechinia caulescens*, and *Caryopteris nepetaefolia* [231,232,233].

The diterpenoid endoperoxide called EBC-325 (**149**) was obtained from an extract of *Croton insularis* [234,235]. The diterpenoid endoperoxide called jungermatrobrunin A (**150**), detected in the liverwort *Jungermannia atrobrunnea*, has an unusual rearrangement-kaurene skeleton with a peroxide bridge [236], and two similar oxygenated diterpenes called triptotins A (**151**) and B (**152**) were found in extracts of the *Tripterygium wilfordii* [237]. The roots of *Jatropha curcas* contained peroxide caniojane (**153**) [238], and another peroxide called steenkrotin B (**154**) was found in ethanol extract of the leaves of *Croton steenkampianus* (Euphorbiaceae), which displayed mild anti-plasmodial activity [239].

Several adamantane type polycyclic polyprenylated acylphloroglucinols (**155**–**164**) possessing an unprecedented *seco*-adamantane architecture combined with a peroxide ring have been isolated and identified from extracts of some plants [240]. Thus, one compound called hypersubone B (**155**) was isolated from the leaves of *Hypericum subsessile* and exhibited significant cytotoxicity against four human cancer lines in vitro, HepG2, Eca109, HeLa, and A549 [241], and hyperisampsins N (**157**) and O (**158**), which exhibited significant cytotoxic activities toward HL-60 cells, were found in the aerial parts of *H. sampsonii* [242]. Structures (**155**–**164**) can be seen in Figure 10, and their biological activity is presented in Table 11.

Peroxysampsones A and B (**156** and **162**) were isolated from the roots of the Chinese medicinal plant *H. sampsonii*, and compound (**156**) showed comparable activity with norfloxacin against a NorA over-expressing multidrug-resistant strain of *Staphylococcus aureus* SA-1199B [243]. Two prenylated benzophenone derivatives, plukenetiones C (**159**) and hydroperoxide (**160**), have been isolated from the fruits of the Barbadian plant *Clusia plukenetii* [244], and otogirinin B (**98**) was detected in *Hypericum erectum* [245]. Garcimultiflorone G (**163**), which shows anti-inflammatory activity, was isolated from the fruits of *Garcinia multiflora* [246], and another polycyclic peroxide called goianone (**164**) was found in fruits extracts of *Clusia rosea* [247].

Unusual polycyclic endoperoxides pregnane glycosides named periplocosides A (**165**), B (**166**) C (**167**), D (**168**), K (**169**), F (**170**), and E (**171**) have been isolated from the antitumor fraction, which was obtained from the CHCl_3_ extract of *Periploca sepium* [248,249,250]. Structures (**165–171**) can be seen in Figure 11, and their biological activity is presented in Table 11.

## 5. Comparison of Biological Activities of Natural Polycyclic Endoperoxides

It is currently accepted that the biological activity of both natural and synthetic compounds depends on their structure [33,251,252]. Despite the activity cliffs observed for some drug-like compounds [253], which can be considered as a violation of this rule, structure-activity relationships (SAR) are widely used in medicinal chemistry for finding and optimization new pharmacological agents [254].

PASS is the first software for in silico estimation of biological activity profiles [33,255], of which the development has been started more than 30 years ago [256]. Its current implementation predicts about 8000 pharmacological effects, molecular mechanisms of action, pharmacological effects, toxicity, side effects, anti-targets, transporters-related interactions, gene expression regulation, and metabolic terms [31]. Due to the utilization of chemical descriptors that reflect the essential features of ligand-target interactions and a robust mathematical approach for analysis of structure-activity relationships, the average accuracy of PASS predictions was 96% [31,252,257,258]. Based on the PASS predictions provided by the appropriate web-service [259], over 29,000 researchers from 104 countries selected the most promising virtually designed molecules for synthesis and determined the optimal directions for testing their biological activity [260,261,262,263,264].

In this study, PASS predictions were used to estimate the general pharmacological potential for the analyzed natural polycyclic endoperoxides. For about 8000 pharmacological effects and molecular mechanisms of action, probabilities of belonging to the class of “actives” Pa, varied from zero to one, were estimated. The higher the Pa value is, the higher the probability of confirming the predicted activity in the experiment. On the other hand, estimated Pa values might be relatively small for some activities if the analyzed molecule is not like the active compounds from the PASS training set. Thus, PASS prediction interpretation requires considering two contradictory issues high probability of activity vs. high structural novelty. The researcher decides which issue is more critical, depending on the task or the project [18,31,35,257,258].

### 5.1. Antiprotozoal Activity of Natural Polycyclic Endoperoxides

Currently, about 120,000 articles have been published that are devoted to antiprotozoal and antiparasitic activities of both natural and synthetic compounds [265,266,267,268,269,270,271].

Analyzing the data obtained with PASS of natural polycyclic endoperoxides and artemisinin and its analogs currently used in medicine, it can be stated that for all polycyclic endoperoxides, antiprotozoal activity is estimated with a Pa from 70 to 99.6%. For some compounds, antiparasitic activity is also estimated, with a Pa from 50 to 88.3%. The antiprotozoal and antiparasitic activities predicted using the PASS are shown in Table 1, Table 2, Table 3, Table 4, Table 5, Table 6, Table 7, Table 8, Table 9, Table 10 and Table 11, and the chemical structures are shown in Figure 1, Figure 2, Figure 3, Figure 4, Figure 5, Figure 6, Figure 7, Figure 8, Figure 9, Figure 10 and Figure 11. A 3D graph of the predicted pharmacological activities of artemisinin (**86**) and its analogs is shown in Figure 12.

Artemisinin and its analogs (both natural and synthetic) are widely used in medical practice and are essential antimalarial treatment components. Figure 12 shows the predicted pharmacological activities of artemisinin and its analogs using PASS, and Figure 13 demonstrates the predicted pharmacological activities of artemisinin.

### 5.2. Antitumor and Other Activities of Natural Polycyclic Endoperoxides

Many natural products exhibit antitumor and related activities and belong to different classes of chemical compounds, such as alkaloids, aromatic and phenolic metabolites, lipids, glycosides, and compounds containing acetylene or epoxy moieties [272,273,274,275,276,277,278]. These compounds also refer to various types of terpenoids, including steroids, triterpenoids, carotenoids, and polycyclic endoperoxides.

More than one million articles and reviews have been devoted to various antitumor and related activities of both natural and synthetic compounds. In an earlier section, we presented and discussed the antitumor activity of polycyclic endoperoxides isolated from various terrestrial and aquatic organisms computed using PASS.

According to the PASS estimates presented in Table 1, Table 2, Table 3, Table 4, Table 5, Table 6, Table 7, Table 8, Table 9, Table 10 and Table 11, many endoperoxides demonstrate antitumor and related activities to varying degrees. However, we are interested in compounds for which such activity is estimated with more than 95% probability. Figure 14 demonstrates natural compounds and their predicted antitumor activity with Pa > 95%.

Some polycyclic endoperoxides, in addition to antiparasitic, antiprotozoal, and antitumor activities, demonstrate other activities with Pa > 90%, which should also be mentioned in this article. This is primarily anti-inflammatory activity. Figure 15 demonstrates such compounds as well as their predicted anti-inflammatory activity. It should also be noted that endoperoxide artemisinin (**86**) and its analogs, and some other compounds, show antifungal activity. Figure 16 demonstrates predicted antifungal activity with Pa> 90%.

## 6. Conclusions

In this review, we presented more than 170 polycyclic endoperoxides isolated from various sources and showed that all endoperoxides demonstrate antiprotozoal activity with varying degrees of reliability, and among them, the artemisinin group and some other compounds are significantly distinguished from of all endoperoxides presented and have a strong antiprotozoal activity. Our data only confirm that the artemisinin group has unique properties, which is why it has been used in medical practice for more than 50 years in the fight against malaria parasites. In addition, the artemisinin group has a high antifungal activity, while some other endoperoxides have a strictly strong anti-inflammatory activity.

Compounds such as (**19**), (**23**), and (**25**) exhibited anti-hypercholesterolemic action, and compounds (**166**) and (**168**) have a strong stimulating effect on the respiratory and vasomotor centers of the brain. However, to confirm the conclusions regarding the *in silico* estimations, more research is required.

## Figures and Tables

**Figure 1 molecules-26-00686-f001:**
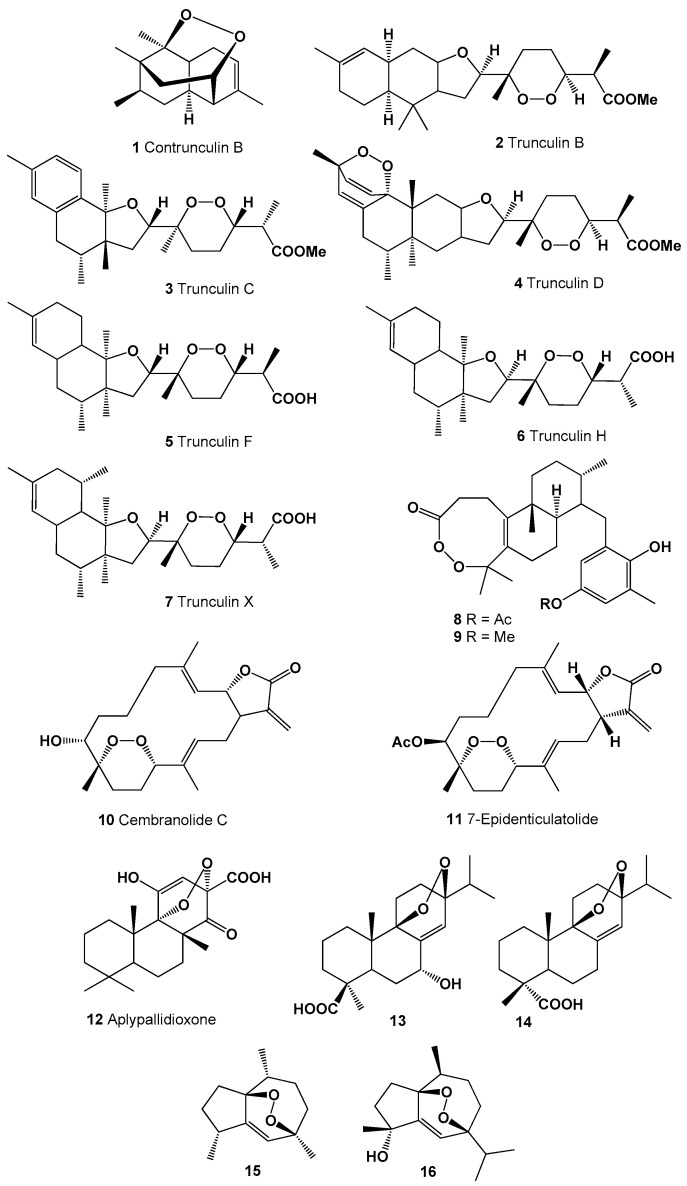
Bioactive polycyclic endoperoxides derived from marine sources.

**Figure 2 molecules-26-00686-f002:**
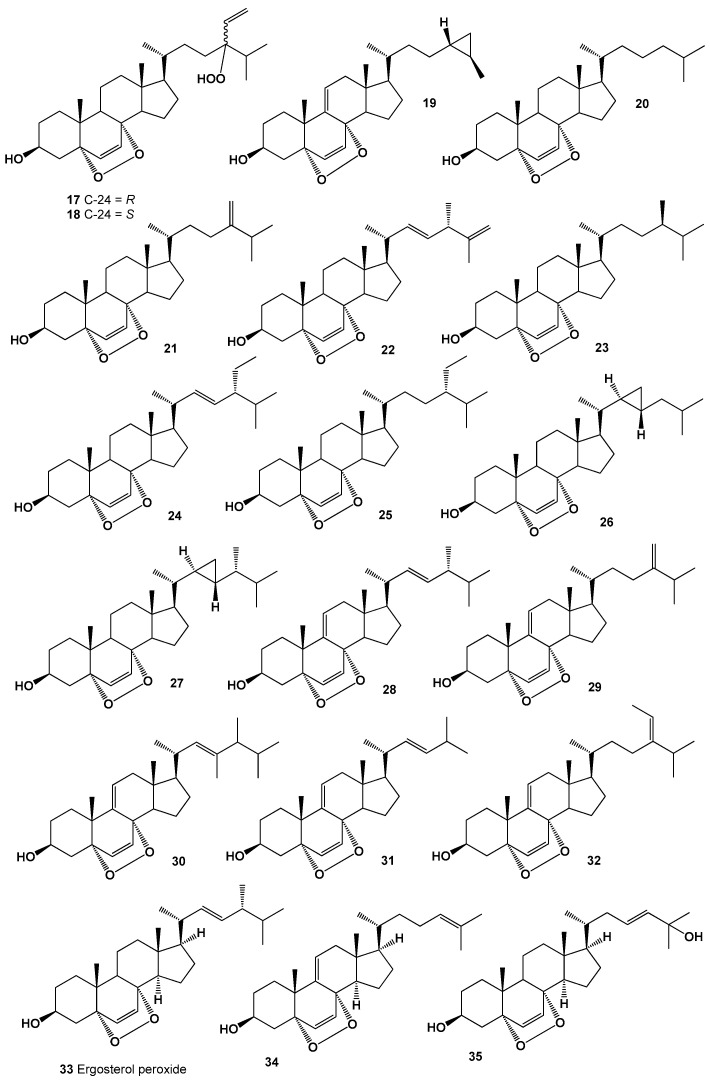
Bioactive polycyclic endoperoxides derived from marine sources.

**Figure 3 molecules-26-00686-f003:**
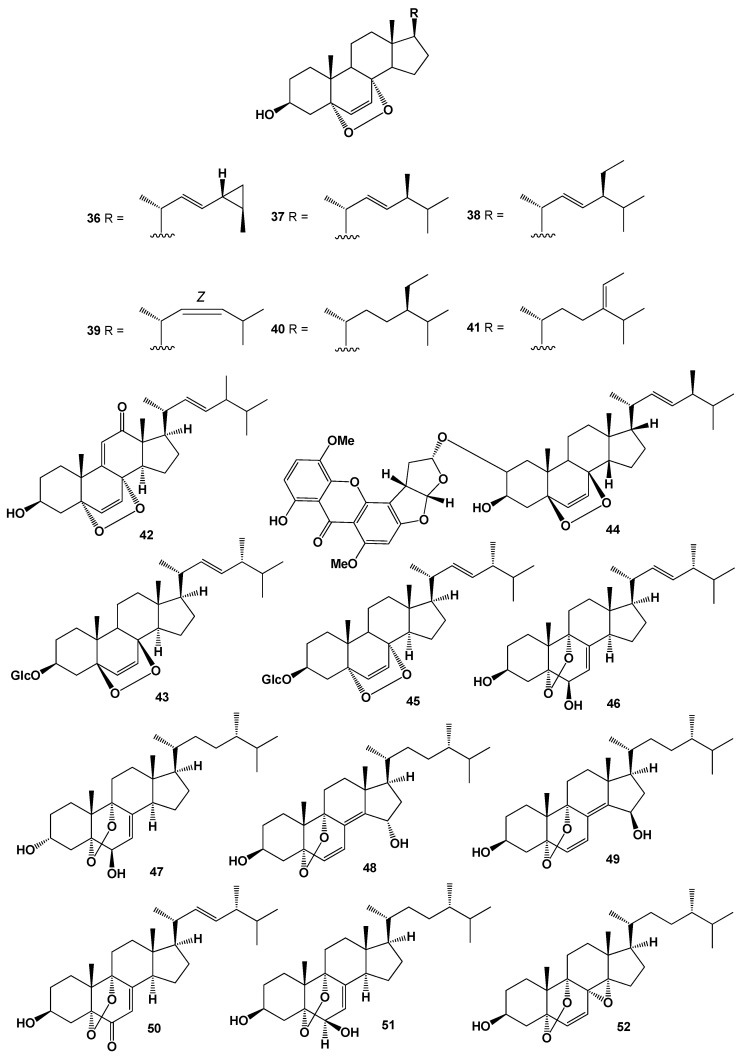
Bioactive polycyclic endoperoxides derived from marine sources and fungi.

**Figure 4 molecules-26-00686-f004:**
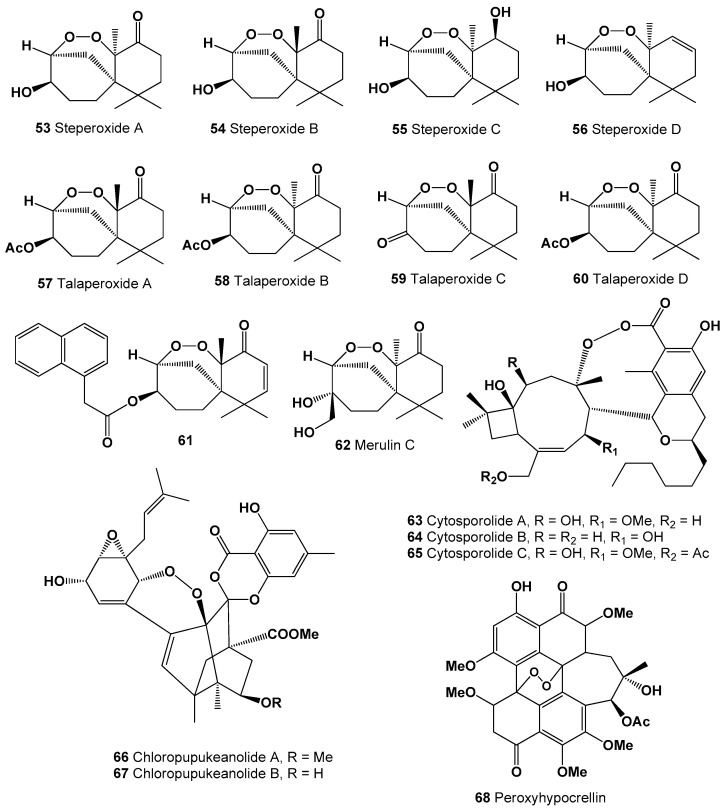
Bioactive polycyclic endoperoxides derived from fungi and fungal endophytes.

**Figure 5 molecules-26-00686-f005:**
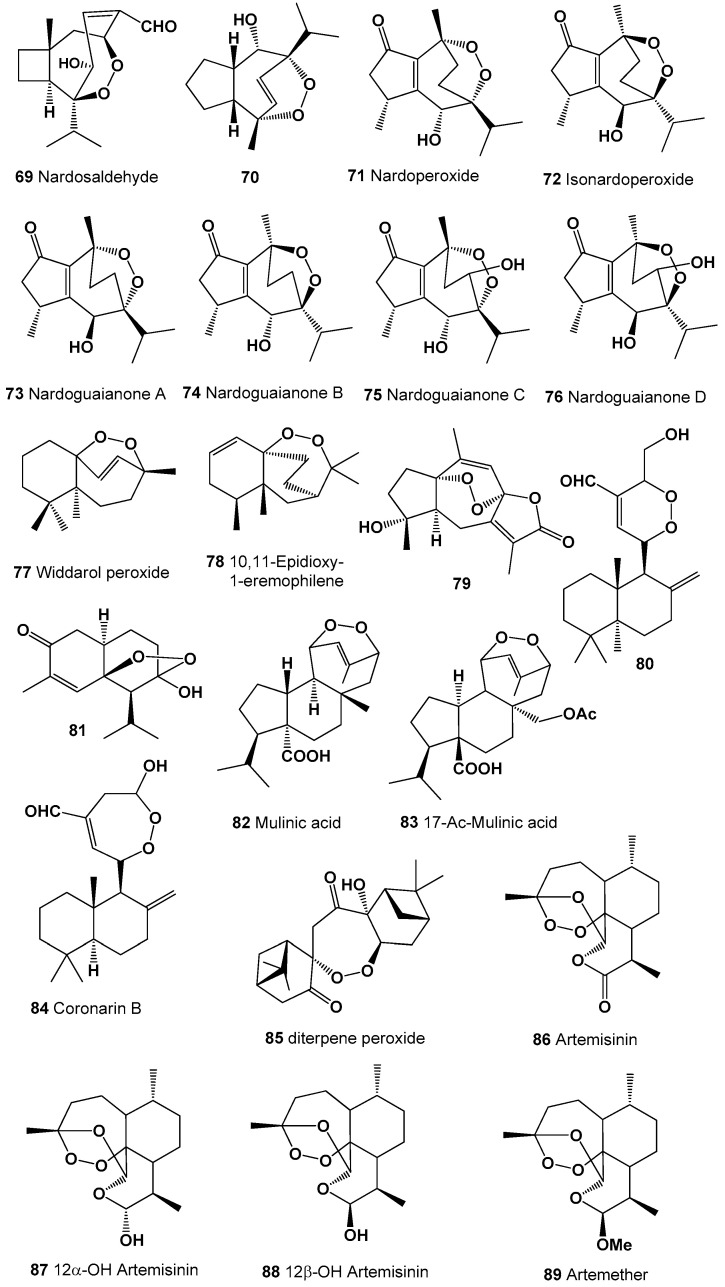
Bioactive polycyclic endoperoxides derived from plants.

**Figure 6 molecules-26-00686-f006:**
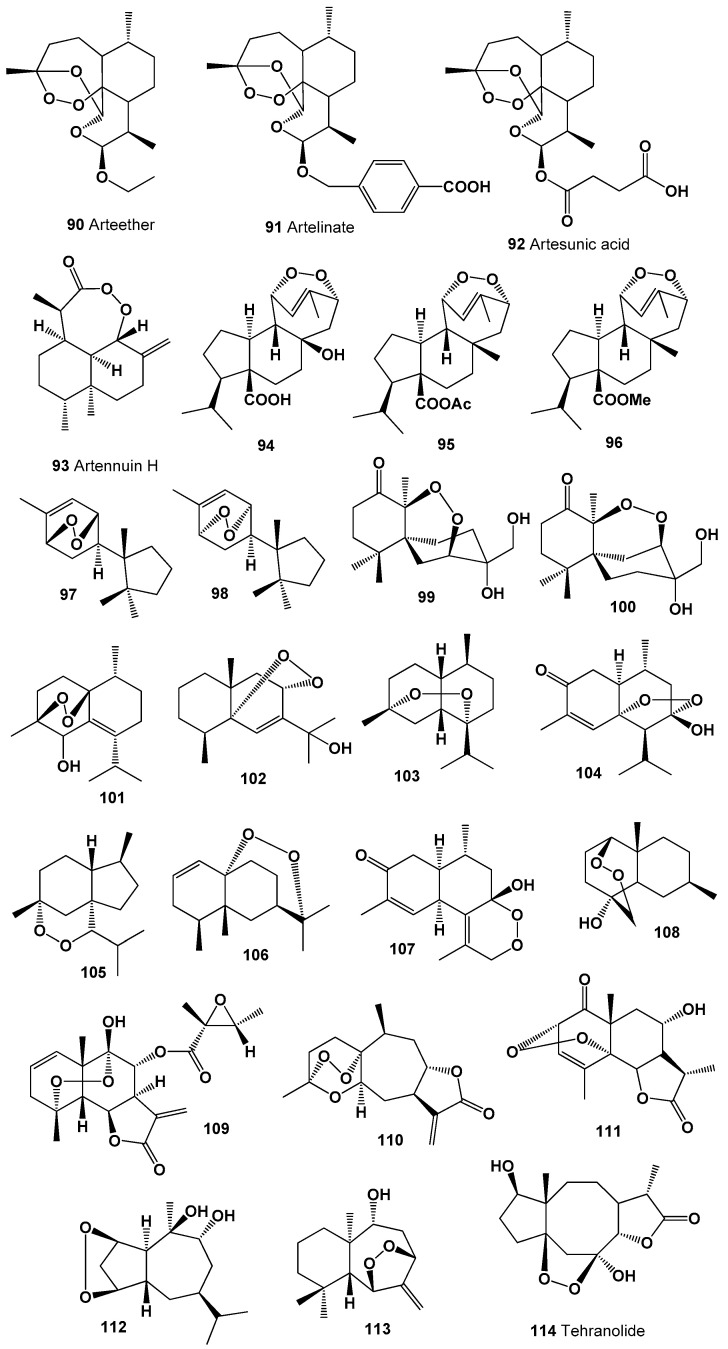
Bioactive polycyclic endoperoxides derived from plants and liverworts.

**Figure 7 molecules-26-00686-f007:**
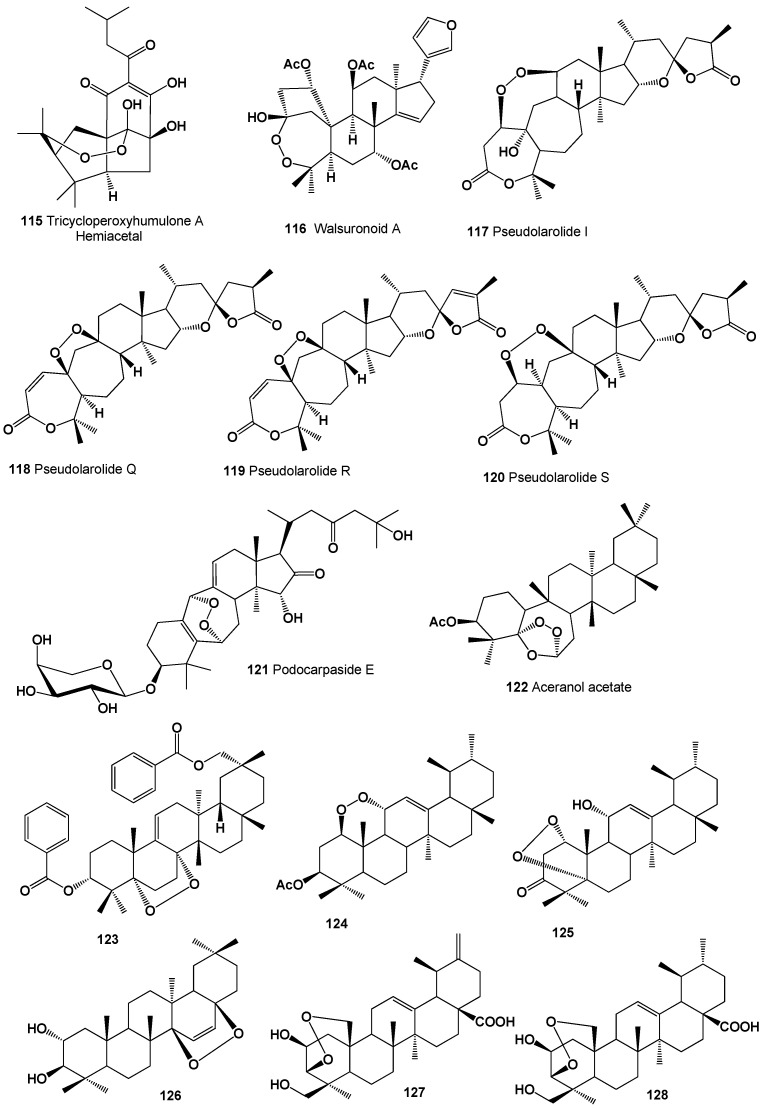
Bioactive polycyclic endoperoxides derived from plants.

**Figure 8 molecules-26-00686-f008:**
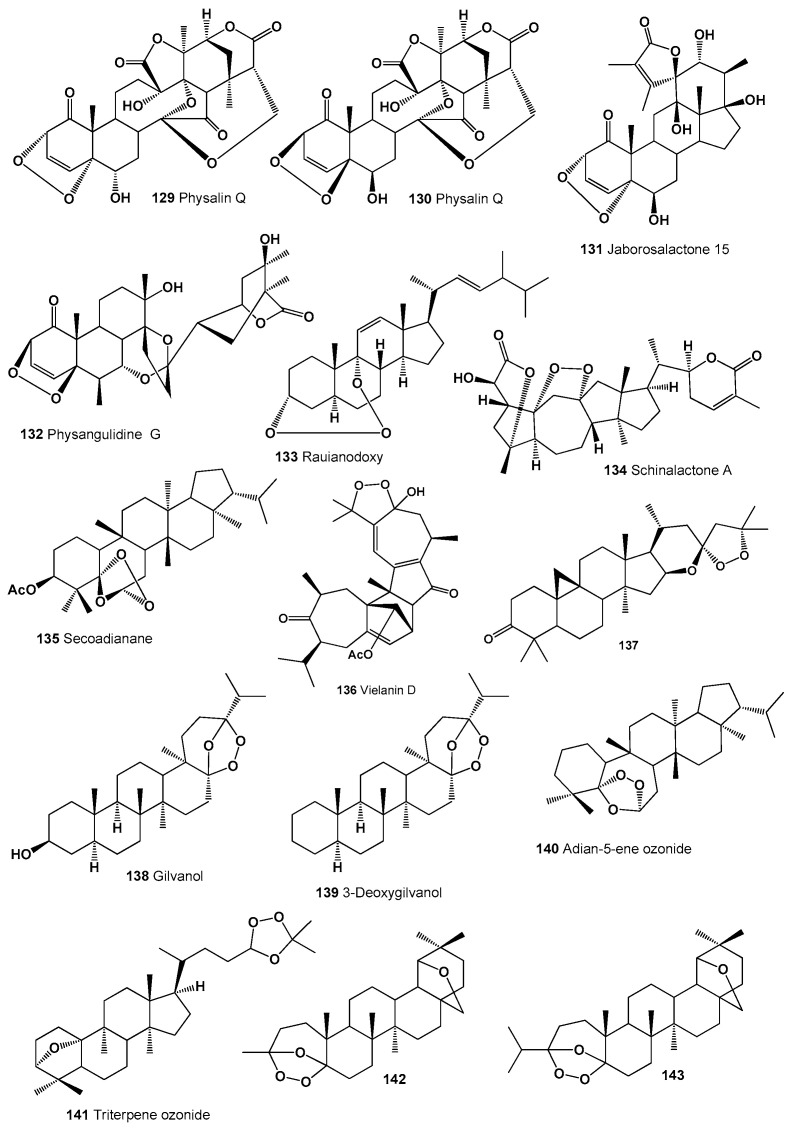
Bioactive polycyclic endoperoxides derived from plants.

**Figure 9 molecules-26-00686-f009:**
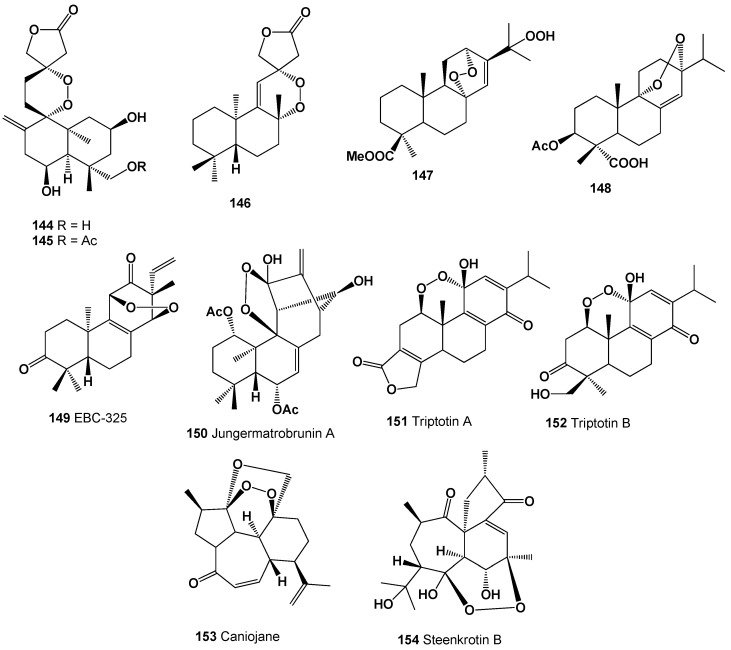
Bioactive polycyclic endoperoxides derived from plants.

**Figure 10 molecules-26-00686-f010:**
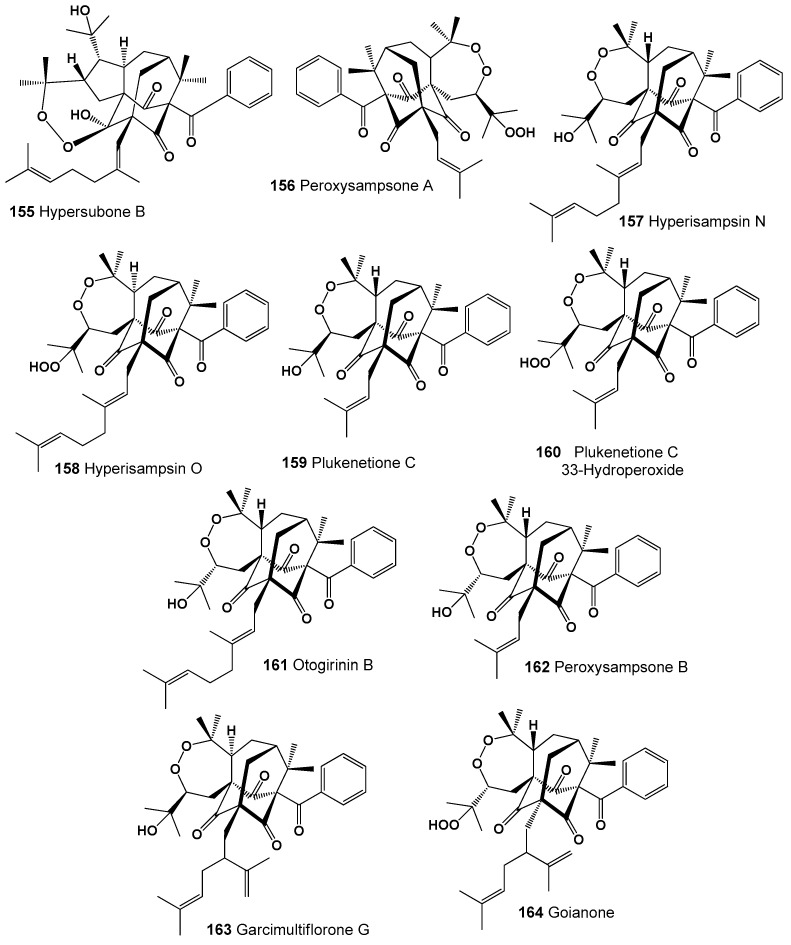
Bioactive adamantane type polycyclic endoperoxides derived from plants.

**Figure 11 molecules-26-00686-f011:**
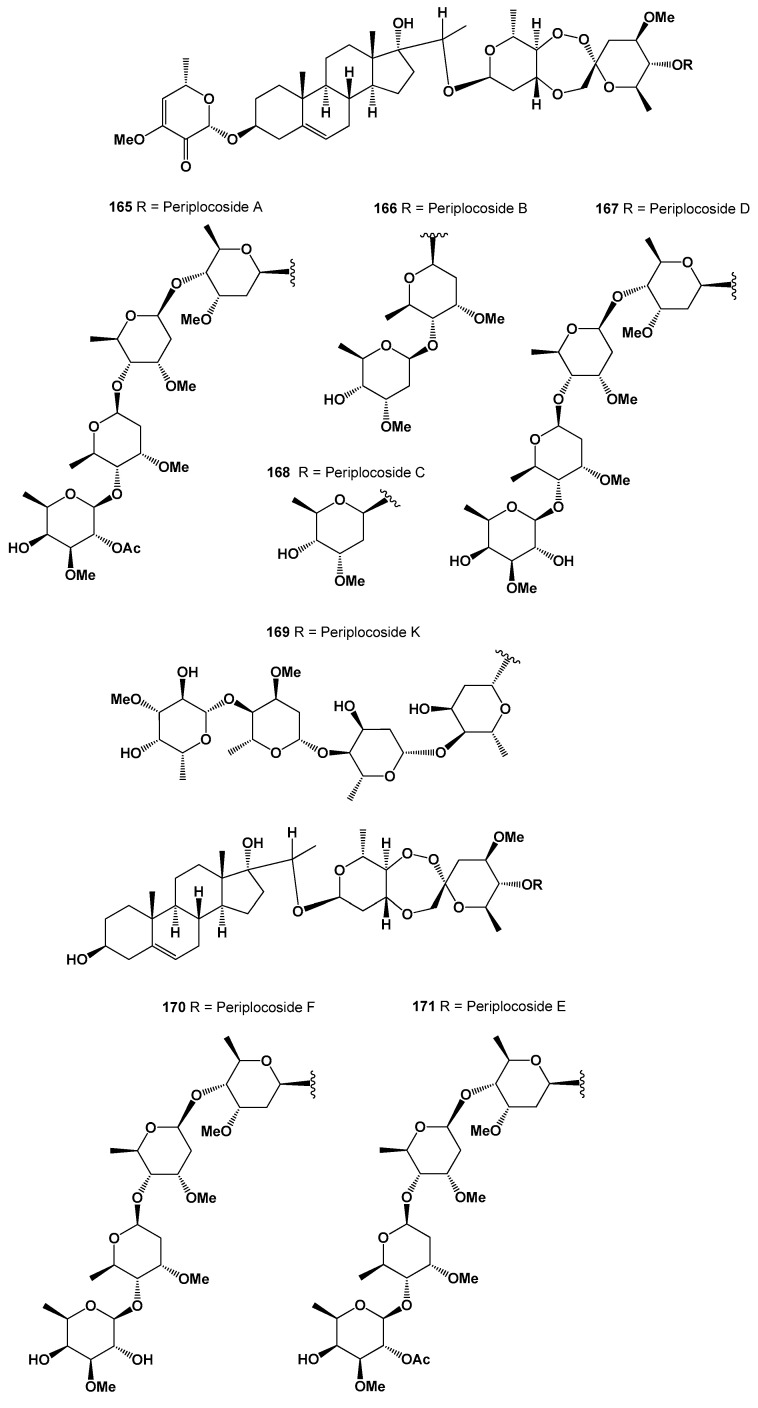
Bioactive steroidal glycosides derived from plants.

**Figure 12 molecules-26-00686-f012:**
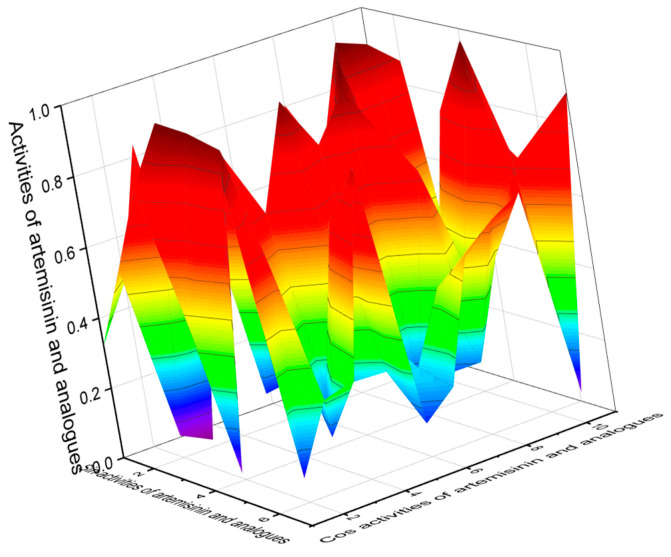
The 3D graph shows the predicted and calculated pharmacological activities of artemisinin (**86**) and its analogs, such as 12α-OH-artemisinin (**87**), 12β-OH- artemisinin (**88**), artemether (**89**), arteether (**90**), artelinate (**91**), and artesunic acid (**92**). According to the PASS data, artemisinin and its analogs (**86**–**92**) show selective activity against obligate intracellular protozoan parasites belonging to the genera *Plasmodium*, *Toxoplasma*, *Leishmania*, and *Coccidia*, which is the main pharmacological activity with a confidence level of more than 90%. In addition, all these endoperoxides show antifungal activity against the opportunistic pathogenic yeasts *Candida* and *Cryptococcus*, as well as anticancer activity for some compounds; the confidence level exceeds 90 percent.

**Figure 13 molecules-26-00686-f013:**
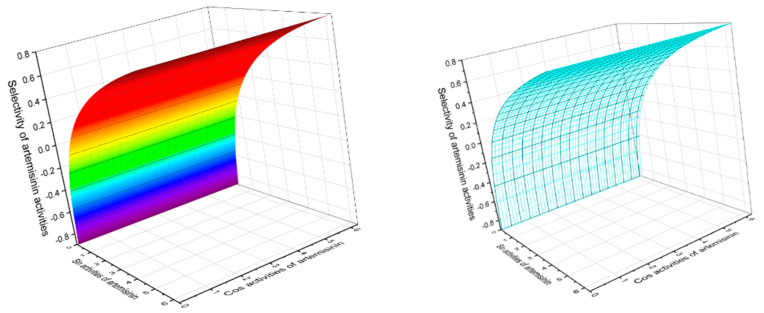
The 3D graph shows the predicted and calculated pharmacological activities of artemisinin (**86**), which was found in 1979 in the extract of the Chinese herb Qinghaosu (*Artemisia annua*). According to PASS data, this endoperoxide demonstrated 16 different activities, with 5 activities having a found confidence of more than 90 percent. Antiprotozoal selective activity of artemisinin against obligate intracellular protozoan parasites belonging to the genera *Plasmodium* (99.5%), *Toxoplasma* (93%), *Leishmania* (92.3%), and *Coccidia* (78%) is the main pharmacological activity. In addition, artemisinin demonstrated strong anti-schistosomal activity (91.1%) against *Schistosoma mansoni*, a human blood fluke parasite. Additionally, artemisinin shows antifungal activity against an opportunistic pathogenic yeast *Candida* (91.5%) and *Cryptococcus* (85.3%), although anticancer activity is found at 80%.

**Figure 14 molecules-26-00686-f014:**
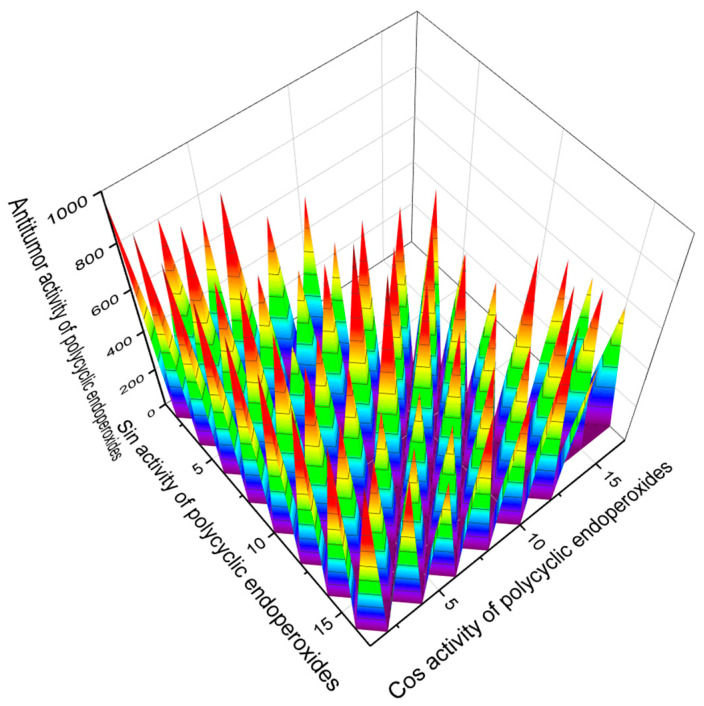
The 3D graph shows the predicted and calculated antitumor activity of selected polycyclic endoperoxides (compound numbers: **11**, **17**, **30**, **33**, **142**, **143**, **164**, and **165**) showing the highest degree of confidence, more than 95%. These polycyclic endoperoxides can be used in clinical medicine as agents with strong antitumor activity.

**Figure 15 molecules-26-00686-f015:**
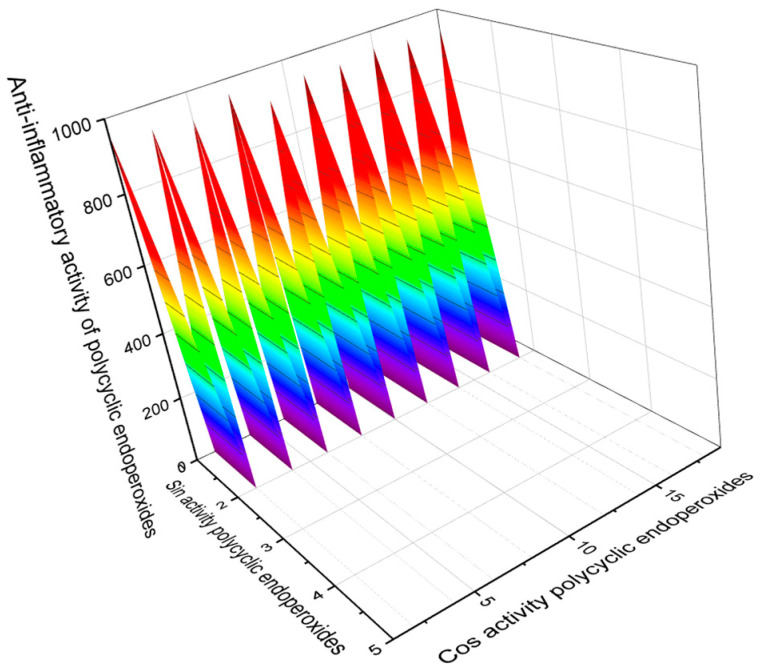
The 3D graph shows the predicted and calculated anti-inflammatory activity of selected polycyclic endoperoxides (compound numbers: **1**, **8**, **9**, **68**, **94**, **95**, **96**, **97**, **98**, **100**, and **113**) showing the highest degree of confidence, i.e., more than 95%. These polycyclic endoperoxides can be used as potential agents with strong anti-inflammatory activity.

**Figure 16 molecules-26-00686-f016:**
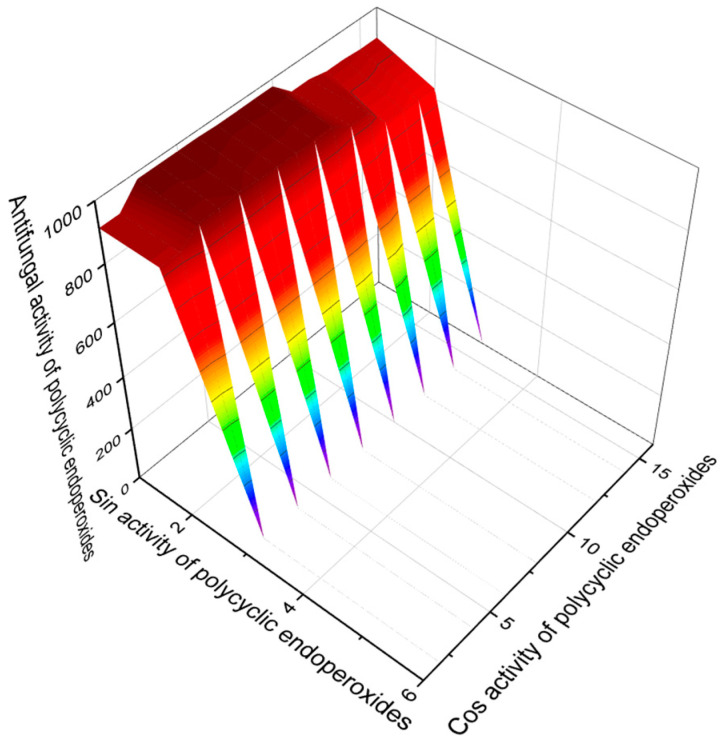
The 3D graph shows the predicted and calculated antifungal activity of selected polycyclic endoperoxides (compound numbers: **86**, **87**, **88**, **89**, **90**, **91**, **92**, **138**, and **139**) polycyclic endoperoxides showing the highest degree of confidence, more than 95%.

**Table 1 molecules-26-00686-t001:** Biological activity of natural polycyclic peroxides derived from marine sources.

No	Antiprotozoal Activity, (Pa) *	Anticancer and Related Activities, (Pa ) *	Additional Biological Activities, (Pa) *
**1**	Antiprotozoal (Plasmodium) (0.941)	Antineoplastic (0.929)	Anti-inflammatory (0.963)
		Apoptosis agonist (0.619)	Antifungal (Candida) (0.638)
**2**	Antiparasitic (0.812)	Antineoplastic (0.722)	Anti-helmintic (0.761)
	Antiprotozoal (Plasmodium) (0.735)	Antimetastatic (0.625)	Antifungal (0.702)
**3**	Antiparasitic (0.812)	Antineoplastic (0.722)	Anti-helmintic (0.761)
	Antiprotozoal (Plasmodium) (0.735)	Antimetastatic (0.625)	Antifungal (0.702)
**4**	Antiparasitic (0.823)	Antineoplastic (0.741)	Anti-helmintic (0.731)
	Antiprotozoal (Plasmodium) (0.714)	Antimetastatic (0.633)	Antifungal (0.673)
**5**	Antiparasitic (0.806)	Antineoplastic (0.755)	Anti-helmintic (0.832)
	Antiprotozoal (Plasmodium) (0.742)	Antimetastatic (0.681)	Antifungal (0.689)
**6**	Antiparasitic (0.786)	Antineoplastic (0.722)	Anti-helmintic (0.774)
	Antiprotozoal (Plasmodium) (0.721)	Antimetastatic (0.625)	Antifungal (0.712)
**7**	Antiparasitic (0.863)	Antineoplastic (0.776)	Anti-helmintic (0.744)
	Antiprotozoal (Plasmodium) (0.716)	Antimetastatic (0.654)	Antifungal (0.731)
**8**	Antiprotozoal (Plasmodium) (0.922)	Antineoplastic (0.913)	Anti-inflammatory (0.937)
**9**	Antiprotozoal (Plasmodium) (0.929)	Antineoplastic (0.929)	Anti-inflammatory (0.929)
**10**	Antiprotozoal (Plasmodium) (0.798)	Antineoplastic (0.975)	Alzheimer’s disease treatment (0.745) Neurodegenerative diseases treatment (0.662)
**11**	Antiprotozoal (Plasmodium) (0.802)	Antineoplastic (0.983)	Alzheimer’s disease treatment (0.722)
**12**	Antiprotozoal (Plasmodium) (0.844)	Antineoplastic (0.921)	Antileukemic (0.599)
		Chemopreventive (0.703)	Immunosuppressant (0.585)
**13**	Antiprotozoal (Plasmodium) (0.835)	Antineoplastic (0.912)	Antileukemic (0.602)
		Chemopreventive (0.658)	Immunosuppressant (0.565)
**14**	Antiprotozoal (Plasmodium) (0.829)	Antineoplastic (0.915)	Antileukemic (0.599)
		Chemopreventive (0.644)	Immunosuppressant (0.602)
**15**	Antiprotozoal (Plasmodium) (0.948)	Antineoplastic (0.835)	Anti-inflammatory (0.576)
	Antiparasitic (0.542)	Antimetastatic (0.635)	
**16**	Antiprotozoal (Plasmodium) (0.964)	Antineoplastic (0.742)	Antileukemic (0.509)
	Antiparasitic (0.642)	Antimetastatic (0.518)	

* Only activities with Pa > 0.5 are shown.

**Table 2 molecules-26-00686-t002:** Biological activity of natural polycyclic peroxides derived from marine sources.

No	Antiprotozoal Activity, (Pa) *	Anticancer and Related Activities, (Pa) *	Additional Biological Activities, (Pa) *
**17**	Antiprotozoal (Plasmodium) (0.925)	Apoptosis agonist (0.961)	Atherosclerosis treatment (0.734)
		Antineoplastic (0.889)	Immunosuppressant (0.721)
**18**	Antiprotozoal (Plasmodium) (0.889)	Apoptosis agonist (0.867)	Anti-inflammatory (0.815)
		Antineoplastic (0.841)	Anti-ulcerative (0.736)
		Antimetastatic (0.611)	
**19**	Antiprotozoal (Plasmodium) (0.838)	Apoptosis agonist (0.977)	Atherosclerosis treatment (0.911)
		Chemopreventive (0.942)	Hypolipemic (0.836)
		Antineoplastic (0.915)	Lipoprotein disorders treatment (0.826)
		Antiparkinsonian, rigidity	Anti-hypercholesterolemic (0.802)
		relieving (0.711)	
		Prostate cancer treatment (0.687)	
**20**	Antiprotozoal (Plasmodium) (0.798)	Apoptosis agonist (0.943)	Atherosclerosis treatment (0.738)
		Antineoplastic (0.767)	Lipoprotein disorders treatment (0.587)
**21**	Antiprotozoal (Plasmodium) (0.694)	Apoptosis agonist (0.961)	Atherosclerosis treatment (0.628)
		Chemopreventive (0.733)	Antifungal (0.635)
		Antineoplastic (0.731)	
**22**	Antiprotozoal (Plasmodium) (0.839)	Apoptosis agonist (0.854)	Anti-inflammatory (0.809)
		Antineoplastic (0.832)	Anti-ulcerative (0.721)
**23**	Antiprotozoal (Plasmodium) (0.871)	Chemopreventive (0.931)	Atherosclerosis treatment (0.907)
		Antineoplastic (0.919)	Anti-hypercholesterolemic (0.788)
**24**	Antiprotozoal (Plasmodium) (0.819)	Apoptosis agonist (0.956)	Atherosclerosis treatment (0.899)
		Chemopreventive (0.933)	Anti-hypercholesterolemic (0.823)
		Antineoplastic (0.905)	
**25**	Antiprotozoal (Plasmodium) (0.822)	Apoptosis agonist (0.966)	Atherosclerosis treatment (0.919)
		Antineoplastic (0.929)	Hypolipemic (0.822)
		Prostate cancer treatment (0.699)	Lipoprotein disorders treatment (0.814)
**26**	Antiprotozoal (Plasmodium) (0.712)	Apoptosis agonist (0.967)	Atherosclerosis treatment (0.635)
		Antineoplastic (0.740)	Antifungal (0.623)
**27**	Antiprotozoal (Plasmodium) (0.713)	Apoptosis agonist (0.966)	Atherosclerosis treatment (0.636)
		Antineoplastic (0.742)	Antifungal (0.624)
**28**	Antiprotozoal (Plasmodium) (0.776)	Apoptosis agonist (0.929)	Atherosclerosis treatment (0.721)
		Antineoplastic (0.758)	Lipoprotein disorders treatment (0.632)
**29**	Antiprotozoal (Plasmodium) (0.778)	Apoptosis agonist (0.922)	Atherosclerosis treatment (0.714)
		Antineoplastic (0.756)	Lipoprotein disorders treatment (0.599)
**30**	Antiprotozoal (Plasmodium) (0.711)	Apoptosis agonist (0.971)	Atherosclerosis treatment (0.623)
		Antineoplastic (0.788)	Antifungal (0.644)
**31**	Antiprotozoal (Plasmodium) (0.699)	Apoptosis agonist (0.954)	Atherosclerosis treatment (0.699)
		Antineoplastic (0.737)	Antifungal (0.676)
**32**	Antiprotozoal (Plasmodium) (0.778)	Apoptosis agonist (0.942)	Atherosclerosis treatment (0.667)
		Antineoplastic (0.732)	Antifungal (0.645)
**33**	Antiprotozoal (Plasmodium) (0.778)	Apoptosis agonist (0.961)	Hypolipemic (0.854)
		Antineoplastic (0.889)	Anti-eczematic (0.812)
		Proliferative diseases treatment (0.522)	Atherosclerosis treatment (0.787)
**34**	Antiprotozoal (Plasmodium) (0.833)	Apoptosis agonist (0.856)	Anti-inflammatory (0.815)
		Antineoplastic (0.838)	Anti-ulcerative (0.729)
**35**	Antiprotozoal (Plasmodium) (0.866)	Apoptosis agonist (0.849)	Anti-inflammatory (0.811)
		Antineoplastic (0.839)	Antifungal (0.677)

* Only activities with Pa > 0.5 are shown.

**Table 3 molecules-26-00686-t003:** Biological activity of natural polycyclic peroxides derived from marine sources and fungi.

No	Antiprotozoal Activity, (Pa) *	Anticancer and Related Activities, (Pa) *	Additional Biological Activities, (Pa) *
**36**	Antiprotozoal (Plasmodium) (0.884)	Antineoplastic (0.859)	Anti-inflammatory (0.881)
		Apoptosis agonist (0.719)	Antifungal (0.644)
**37**	Antiprotozoal (Plasmodium) (0.816)	Antineoplastic (0.725)	Analgesic (0.812)
	Antiparasitic (0.806)	Antimetastatic (0.625)	Antifungal (0.745)
**38**	Antiprotozoal (Plasmodium) (0.789)	Antineoplastic (0.721)	Analgesic (0.806)
	Antiparasitic (0.807)	Antimetastatic (0.625)	Antifungal (0.730)
**39**	Antiprotozoal (Plasmodium) (0.765)	Antineoplastic (0.732)	Antileukemic (0.729)
	Antiparasitic (0.763)	Antimetastatic (0.633)	Antifungal (0.670)
**40**	Antiprotozoal (Plasmodium) (0.778)	Antineoplastic (0.726)	Antileukemic (0.717)
	Antiparasitic (0.745)	Antimetastatic (0.681)	Antifungal (0.669)
**41**	Antiparasitic (0.814)	Antineoplastic (0.734)	Anti-helmintic (0.765)
	Antiprotozoal (Plasmodium) (0.744)	Antimetastatic (0.625)	Antifungal (0.708)
**42**	Antiparasitic (0.863)	Antineoplastic (0.799)	Anti-helmintic (0.731)
	Antiprotozoal (Plasmodium) (0.716)	Antimetastatic (0.689)	Antifungal (0.705)
**43**	Antiprotozoal (Plasmodium) (0.882)	Antineoplastic (0.824)	Anti-inflammatory (0.821)
**44**	Antiprotozoal (Plasmodium) (0.702)	Apoptosis agonist (0.910)	Anti-inflammatory (0.686)
		Antineoplastic (0.782)	Antileukemic (0.659)
**45**	Antiprotozoal (Plasmodium) (0.898)	Antineoplastic (0.856)	Alzheimer’s disease treatment (0.732)
**46**	Antiprotozoal (Plasmodium) (0.775)	Antineoplastic (0.868)	Alzheimer’s disease treatment (0.698)
**47**	Antiprotozoal (Plasmodium) (0.844)	Antineoplastic (0.843)	Antifungal (0.659)
		Chemopreventive (0.712)	Immunosuppressant (0.582)
**48**	Antiprotozoal (Plasmodium) (0.835)	Antineoplastic (0.823)	Antifungal (0.662)
		Chemopreventive (0.679)	Immunosuppressant (0.565)
**49**	Antiprotozoal (Plasmodium) (0.829)	Antineoplastic (0.818)	Antileukemic (0.645)
		Chemopreventive (0.644)	Immunosuppressant (0.602)
**50**	Antiprotozoal (Plasmodium) (0.743)	Antineoplastic (0.788)	Antifungal (0.670)
	Antiparasitic (0.671)	Antimetastatic (0.603)	Anti-inflammatory (0.656)
**51**	Antiprotozoal (Plasmodium) (0.752)	Antineoplastic (0.859)	Antifungal (0.670)
		Prostate cancer treatment (0.655)	Anti-inflammatory (0.661)
**52**	Antiprotozoal (Plasmodium) (0.752)	Antineoplastic (0.859)	Analgesic (0.843)

* Only activities with Pa > 0.5 are shown.

**Table 4 molecules-26-00686-t004:** Bioactivity of natural polycyclic peroxides derived from fungi and fungal endophytes.

No	Antiprotozoal Activity, (Pa) *	Anticancer and Related Activities, (Pa) *	Additional Biological Activities, (Pa) *
**53**	Antiprotozoal (Plasmodium) (0.911)	Antineoplastic enhancer (0.825)	Antifungal (0.688)
	Antiparasitic (0.643)	Antineoplastic (0.794)	Anti-inflammatory (0.661)
**54**	Antiprotozoal (Plasmodium) (0.916)	Antineoplastic (0.833)	Antifungal (0.676)
	Antiparasitic (0.650)	Apoptosis agonist (0.710)	Anti-inflammatory (0.661)
**55**	Antiprotozoal (Plasmodium) (0.919)		Antineoplastic (0.839)
**56**	Antiprotozoal (Plasmodium) (0.904)	Antineoplastic (0.814)	Antifungal (0.646)
	Antiparasitic (0.649)	Apoptosis agonist (0.670)	Anti-inflammatory (0.589)
**57**	Antiprotozoal (Plasmodium) (0.913)	Antineoplastic (0.795)	Antifungal (0.688)
		Apoptosis agonist (0.760)	Anti-inflammatory (0.612)
**58**	Antiprotozoal (Plasmodium) (0.902)	Antineoplastic enhancer (0.825)	Antifungal (0.646)
	Antiparasitic (0.666)	Antineoplastic (0.794)	Anti-inflammatory (0.611)
**59**	Antiprotozoal (Plasmodium) (0.923)	Antineoplastic (0.865)	Antifungal (0.671)
**60**	Antiprotozoal (Plasmodium) (0.908)	Antineoplastic enhancer (0.816)	Antifungal (0.677)
	Antiparasitic (0.652)	Antineoplastic (0.799)	Anti-inflammatory (0.622)
**61**	Antiprotozoal (Plasmodium) (0.920)	Antineoplastic (0.825)	Antifungal (0.721)
**62**	Antiprotozoal (Plasmodium) (0.935)	Antineoplastic (0.716)	Antifungal (0.709)
		Antineoplastic (renal cancer) (0.598)	
**63**	Antiprotozoal (Plasmodium) (0.839)	Antineoplastic (0.756)	Antifungal (0.705)
	Antiparasitic (0.780)	Antineoplastic (renal cancer) (0.592)	
**64**	Antiprotozoal (Plasmodium) (0.836)	Antineoplastic (0.758)	Antifungal (0.711)
**65**	Antiprotozoal (Plasmodium) (0.835)	Antineoplastic (0.756)	Antifungal (0.705)
**66**	Antiprotozoal (Plasmodium) (0.877)	Antineoplastic (0.848)	Antiviral (0.768)
**67**	Antiprotozoal (Plasmodium) (0.877)	Antineoplastic (0.848)	Antiviral (0.768)
**68**	Antiprotozoal (Plasmodium) (0.938)	Antineoplastic (0.912)	Anti-inflammatory (0.908)

* Only activities with Pa > 0.5 are shown.

**Table 5 molecules-26-00686-t005:** Biological activity of the natural polycyclic peroxides derived from plants.

No	Antiprotozoal Activity, (Pa) *	Anticancer and Related Activities, (Pa) *	Additional Biological Activities, (Pa) *
**69**	Antiprotozoal (Plasmodium) (0.930)	Antineoplastic (0.674)	Phobic disorders treatment (0.604)
		Antimetastatic (0.536)	Ovulation inhibitor (0.550)
**70**	Antiprotozoal (Plasmodium) (0.756)	Antineoplastic (0.787)	Analgesic (0.883)
	Antiparasitic (0.662)	Antimetastatic (0.591)	
**71**	Antiprotozoal (Plasmodium) (0.729)	Antineoplastic (0.788)	Analgesic (0.883)
	Antiparasitic (0.662)	Antimetastatic (0.591)	Antileukemic (0.564)
**72**	Antiprotozoal (Plasmodium) (0.743)	Antineoplastic (0.769)	Analgesic (0.883)
		Antimetastatic (0.591)	Antileukemic (0.564)
**73**	Antiprotozoal (Plasmodium) (0.755)	Antineoplastic (0.801)	Analgesic (0.883)
	Antiparasitic (0.662)	Antimetastatic (0.591)	Antileukemic (0.564)
**74**	Antiprotozoal (Plasmodium) (0.722)	Antineoplastic (0.855)	Analgesic (0.843)
	Antiparasitic (0.510)	Prostate cancer treatment (0.641)	Anti-inflammatory (0.648)
**75**	Antiprotozoal (Plasmodium) (0.739)	Antineoplastic (0.855)	Analgesic (0.843)
	Antiparasitic (0.510)	Prostate cancer treatment (0.641)	Antileukemic (0.513)
		Antimetastatic (0.517)	Antibacterial (0.503)
**76**	Antiprotozoal (Plasmodium) (0.964)	Apoptosis agonist (0.862)	Antifungal (0.538)
		Antineoplastic (0.694)	Antiviral (Arbovirus) (0.536)
**77**	Antiprotozoal (Plasmodium) (0.954)	Apoptosis agonist (0.910)	Atherosclerosis treatment (0.520)
	Antiparasitic (0.553)	Antineoplastic (0.768)	
		Antimetastatic (0.587)	
**78**	Antiprotozoal (Plasmodium) (0.805)	Antineoplastic (0.949)	Anti-inflammatory (0.924)
		Apoptosis agonist (0.797)	Antifungal (0.703)
		Antimetastatic (0.505)	
**79**	Antiprotozoal (Plasmodium) (0.855)	Antineoplastic (0.582)	Immunosuppressant (0.706)
**80**	Antiprotozoal (Plasmodium) (0.900)	Antineoplastic (0.873)	Anti-psoriatic (0.630)
**81**	Antiprotozoal (Plasmodium) (0.964)	Antineoplastic (0.602)	Phobic disorders treatment (0.725)

* Only activities with Pa > 0.5 are shown.

**Table 6 molecules-26-00686-t006:** Biological activity of the natural polycyclic peroxides derived from plants.

No	Antiprotozoal Activity, (Pa) *	Anticancer and Related Activities, (Pa) *	Additional Biological Activities, (Pa) *
**82**	Antiprotozoal (Plasmodium) (0.868)	Antineoplastic (0.887)	Anti-inflammatory (0.946)
**83**	Antiprotozoal (Plasmodium) (0.809)	Antineoplastic (0.813)	Anti-inflammatory (0.898)
**84**	Antiprotozoal (Plasmodium) (0.882)	Apoptosis agonist (0.948)	Anti-inflammatory (0.867)
		Antineoplastic (0.921)	Antifungal (0.789)
**85**	Antiprotozoal (Plasmodium) (0.912)	Antineoplastic (0.813)	Cardiotonic (0.939)
			Cardiovascular analeptic (0.660)
**86**	Antiprotozoal (Plasmodium) (0.996)	Antineoplastic (0.797)	Antifungal (Candida) (0.915)
	Antiprotozoal (Toxoplasma) (0.930)	Apoptosis agonist (0.787)	Anti-schistosome (0.911)
	Antiprotozoal (Leishmania) (0.923)	DNA synthesis inhibitor (0.747)	Antifungal (Cryptococcus) (0.853)
	Antiparasitic (0.869)	Immunosuppressant (0.720)	Diuretic (0.837)
	Antiprotozoal (Coccidia) (0.780)		Antifungal (0.827)
**87**	Antiprotozoal (Plasmodium) (0.996)	Apoptosis agonist (0.919)	Antifungal (Candida) (0.979)
	Antiprotozoal (Leishmania) (0.966)	Antineoplastic (0.847)	Anti-schistosome (0.961)
	Antiprotozoal (Toxoplasma) (0.918) Antiparasitic (0.883)	DNA synthesis inhibitor (0.644)	Antifungal (Cryptococcus) (0.955)
	Antiprotozoal (Coccidia) (0.794)		Antifungal (0.846)
**88**	Antiprotozoal (Plasmodium) (0.996)	Apoptosis agonist (0.919)	Antifungal (Candida) (0.979)
	Antiprotozoal (Leishmania) (0.966)	Antineoplastic (0.847)	Anti-schistosome (0.961)
	Antiprotozoal (Toxoplasma) (0.918)	DNA synthesis inhibitor (0.644)	Antifungal (Cryptococcus) (0.955)
	Antiparasitic (0.883)		Antifungal (0.846)
	Antiprotozoal (Coccidia) (0.794)		Angiogenesis inhibitor (0.738)
**89**	Antiprotozoal (Plasmodium) (0.996)	Apoptosis agonist (0.890)	Antifungal (Candida) (0.976)
	Antiprotozoal (Leishmania) (0.949)	Antineoplastic (0.820)	Anti-schistosome (0.975)
	Antiprotozoal (Toxoplasma) (0.928)	Immunosuppressant (0.704)	Antifungal (Cryptococcus) (0.953)
	Antiparasitic (0.880)	DNA synthesis inhibitor (0.590)	Antifungal (0.828)
	Antiprotozoal (Coccidia) (0.792)		Antifungal (Aspergillus) (0.627)
**90**	Antiprotozoal (Plasmodium) (0.996)	Apoptosis agonist (0.866)	Antifungal (Candida) (0.977)
	Antiprotozoal (Leishmania) (0.957)	Antineoplastic (0.793)	Anti-schistosome (0.970)
	Antiprotozoal (Toxoplasma) (0.918)	DNA synthesis inhibitor (0.545)	Antifungal (Cryptococcus) (0.950)
	Antiparasitic (0.880)		Antifungal (0.832)
	Antiprotozoal (Coccidia) (0.818)		Antifungal (Aspergillus) (0.761)
**91**	Antiprotozoal (Plasmodium) (0.982)	Apoptosis agonist (0.787)	Antifungal (Candida) (0.921)
	Antiprotozoal (Leishmania) (0.966)	Antineoplastic (0.755)	Anti-schistosome (0.915)
	Antiparasitic (0.876)	DNA synthesis inhibitor (0.592)	Antifungal (0.849)
	Antiprotozoal (Toxoplasma) (0.875)		Antifungal (Cryptococcus) (0.749)
	Antiprotozoal (Coccidia) (0.649)		Antifungal (Aspergillus) (0.631)
			Antiviral (CMV) (0.603)
**92**	Antiprotozoal (Plasmodium) (0.990)	Apoptosis agonist (0.884)	Anti-schistosome (0.960)
	Antiprotozoal (Leishmania) (0.929)	Antineoplastic (0.828)	Antifungal (Candida) (0.942)
	Antiprotozoal (Toxoplasma) (0.899)	DNA synthesis inhibitor (0.607)	Antifungal (0.868)
	Antiparasitic (0.886)		Antifungal (Cryptococcus) (0.825)
	Antiprotozoal (Coccidia) (0.689)		Antiviral (CMV) (0.668)
			Antifungal (Aspergillus) (0.607)
**93**	Antiprotozoal (Plasmodium) (0.860)	Antineoplastic (0.883)	Anti-eczematic (0.934)
		Prostate disorders treatment (0.675)	Anti-inflammatory (0.819)
			Cardiovascular analeptic (0.733)
			Anti-psoriatic (0.690)

* Only activities with Pa > 0.5 are shown.

**Table 7 molecules-26-00686-t007:** Biological activity of natural polycyclic peroxides derived from plants.

No	Antiprotozoal Activity, (Pa) *	Anticancer and Related Activities, (Pa) *	Additional Biological Activities, (Pa) *
**94**	Antiprotozoal (Plasmodium) (0.868)	Antineoplastic (0.887)	Anti-inflammatory (0.946)
**95**	Antiprotozoal (Plasmodium) (0.874)	Antineoplastic (0.870)	Anti-inflammatory (0.941)
**96**	Antiprotozoal (Plasmodium) (0.889)	Antineoplastic (0.769)	Anti-inflammatory (0.918)
**97**	Antiprotozoal (Plasmodium) (0.936)	Antineoplastic (0.932)	Anti-inflammatory (0.958)
		Apoptosis agonist (0.617)	Antifungal (Candida) (0.630)
**98**	Antiprotozoal (Plasmodium) (0.936)	Antineoplastic (0.932)	Anti-inflammatory (0.958)
		Apoptosis agonist (0.617)	Antifungal (Candida) (0.630)
**99**	Antiprotozoal (Plasmodium) (0.928)	Antineoplastic (0.681)	Anti-inflammatory (0.544)
**100**	Antiprotozoal (Plasmodium) (0.916)	Antineoplastic (0.854)	Anti-inflammatory (0.945)
**101**	Antiprotozoal (Plasmodium) (0.879)	Antineoplastic (0.866)	Anti-inflammatory (0.934)
	Antiparasitic (0.649)	Antimetastatic (0.623)	Anti-helminthic (0.609)
**102**	Antiprotozoal (Plasmodium) (0.965)	Antineoplastic (0.792)	Carminative (0.652)
	Antiparasitic (0.576)	Antimetastatic (0.584)	
**103**	Antiprotozoal (Plasmodium) (0.954)	Apoptosis agonist (0.565)	Carminative (0.832)
**104**	Antiprotozoal (Plasmodium) (0.956)	Antineoplastic (0.670)	Anti-eczematic (0.700)
	Antiparasitic (0.574)	Antimetastatic (0.587)	Antifungal (0.593)
**105**	Antiprotozoal (Plasmodium) (0.959)	Antineoplastic (0.678)	Anti-eczematic (0.711)
	Antiparasitic (0.581)	Antimetastatic (0.588)	Antifungal (0.599)
**106**	Antiprotozoal (Plasmodium) (0.938)	Antineoplastic (sarcoma) (0.734)	Carminative (0.812)
**107**	Antiprotozoal (Plasmodium) (0.945)	Antineoplastic (sarcoma) (0.529)	Anti-eczematic (0.715)
**108**	Antiprotozoal (Plasmodium) (0.881)	Antineoplastic (0.699)	Anti-eczematic (0.734)
**109**	Antiprotozoal (Plasmodium) (0.884)	Antineoplastic (0.862)	Anti-eczematic (0.861)
	Antiparasitic (0.672)	Apoptosis agonist (0.795)	Anti-inflammatory (0.679)
**110**	Antiprotozoal (Plasmodium) (0.967)	Antineoplastic (0.911)	Anti-eczematic (0.836)
	Antiparasitic (0.811)	Apoptosis agonist (0.883)	Antifungal (0.812)
	Antiprotozoal (Leishmania) (0.731)	DNA synthesis inhibitor (0.652)	Antibacterial (0.667)
**111**	Antiprotozoal (Plasmodium) (0.889)	Antineoplastic (0.769)	Angiogenesis stimulant (0.644)
**112**	Antiprotozoal (Plasmodium) (0.917)	Antineoplastic (0.797)	Carminative (0.724)
		Prostate cancer treatment (0.650)	Anti-inflammatory (0.697)
**113**	Antiprotozoal (Plasmodium) (0.752)	Antineoplastic (0.946)	Anti-inflammatory (0.949)
		Apoptosis agonist (0.782)	Anti-eczematic (0.896)
**114**	Antiprotozoal (Plasmodium) (0.925)	Antineoplastic (0.914)	Anti-eczematic (0.851)
	Antiparasitic (0.741)		Anti-helmintic (0.702)

* Only activities with Pa > 0.5 are shown.

**Table 8 molecules-26-00686-t008:** Biological activity of natural polycyclic peroxides derived from plants.

No	Antiprotozoal Activity, (Pa) *	Anticancer and Related Activities, (Pa) *	Additional Biological Activities, (Pa) *
**115**	Antiprotozoal (Plasmodium) (0.920)	Antineoplastic (0.719)	Allergic conjunctivitis treatment (0.597)
		Apoptosis agonist (0.716)	
**116**	Antiprotozoal (Plasmodium) (0.904)	Antineoplastic (0.752)	Anti-inflammatory (0.815)
		Apoptosis agonist (0.656)	Antifungal (0.533)
**117**	Antiprotozoal (Plasmodium) (0.886)	Antineoplastic (0.899)	Antifungal (0.807)
	Antiparasitic (0.548)	Apoptosis agonist (0.852)	Antimitotic (0.690)
**118**	Antiprotozoal (Plasmodium) (0.891)	Antineoplastic (0.902)	Antifungal (0.854)
	Antiparasitic (0.603)	Apoptosis agonist (0.833)	Antimitotic (0.702)
**119**	Antiprotozoal (Plasmodium) (0.877)	Antineoplastic (0.904)	Antifungal (0.836)
	Antiparasitic (0.567)	Apoptosis agonist (0.834)	Antimitotic (0.721)
**120**	Antiprotozoal (Plasmodium) (0.878)	Antineoplastic (0.879)	Antifungal (0.823)
	Antiparasitic (0.601)	Apoptosis agonist (0.821)	Antimitotic (0.704)
**121**	Antiprotozoal (Plasmodium) (0.734)	Antineoplastic (0.828)	Anti-psoriatic (0.607)
		Chemopreventive (0.785)	Anti-eczematic (0.546)
**122**	Antiprotozoal (Plasmodium) (0.955)	apoptosis agonist (0.783)	Anti-inflammatory (0.731)
		Antineoplastic (0.762)	Lipid metabolism regulator (0.617)
**123**	Antiprotozoal (Plasmodium) (0.702)	Apoptosis agonist (0.910)	Anti-inflammatory (0.686)
		Antineoplastic (0.782)	Lipid metabolism regulator (0.631)
**124**	Antiprotozoal (Plasmodium) (0.869)	Apoptosis agonist (0.853)	Anti-inflammatory (0.869)
	Antiprotozoal (Leishmania) (0.582)	Antineoplastic (0.804)	Antifungal (0.707)
**125**	Antiprotozoal (Plasmodium) (0.848)	Antineoplastic (0.821)	Anti-inflammatory (0.899)
**126**	Antiprotozoal (Plasmodium) (0.835)	Apoptosis agonist (0.919)	Anti-inflammatory (0.858)
		Antineoplastic (0.842)	Diuretic (0.748)
**127**	Antiprotozoal (Plasmodium) (0.891)	Antineoplastic (0.874)	Hepatoprotectant (0.838)
		Apoptosis agonist (0.871)	Antifungal (0.716)
**128**	Antiprotozoal (Plasmodium) (0.891)	Antineoplastic (0.874)	Hepatoprotectant (0.838)
		Apoptosis agonist (0.871)	Antifungal (0.716)

* Only activities with Pa > 0.5 are shown.

**Table 9 molecules-26-00686-t009:** Biological activity of natural polycyclic peroxides derived from plants.

No	Antiprotozoal Activity, (Pa) *	Anticancer and Related Activities, (Pa) *	Additional Biological Activities, (Pa) *
**129**	Antiprotozoal (0.969)	Antineoplastic (0.812)	Immunosuppressant (0.586)
	Antiprotozoal (Plasmodium) (0.966)	Antimetastatic (0.504)	Antifungal (0.521)
**130**	Antiprotozoal (0.967)	Antineoplastic (0.819)	Antibacterial (0.657)
	Antiprotozoal (Plasmodium) (0.966)	Antimetastatic (0.504)	Antifungal (0.521)
**131**	Antiprotozoal (0.837)	Antineoplastic (0.788)	Antifungal (0.609)
	Antiprotozoal (Plasmodium) (0.820)		Antibacterial (0.557)
**132**	Antiprotozoal (Plasmodium) (0.883)	Antineoplastic (0.890)	Antifungal (0.666)
**133**	Antiprotozoal (Plasmodium) (0.846)	Apoptosis agonist (0.934)	Hypolipemic (0.820)
		Antineoplastic (0.890)	Anti-hypercholesterolemic (0.608)
		Prostate cancer treatment (0.636)	Atherosclerosis treatment (0.679)
**134**	Antiprotozoal (Plasmodium) (0.894)	Antineoplastic (0.875)	Antifungal (0.703)
**135**	Antiprotozoal (Plasmodium) (0.952)	Antineoplastic (0.756)	Anti-eczematic (0.863)
		Apoptosis agonist (0.689)	Anti-psoriatic (0.640)
**136**	Antiprotozoal (Plasmodium) (0.782)	Antineoplastic (0.879)	Anti-eczematic (0.684)
	Antiprotozoal (0.776)	Antineoplastic (sarcoma) (0.671)	Anti-inflammatory (0.681)
		Antineoplastic (renal cancer) (0.615)	
**137**	Antiprotozoal (Plasmodium) (0.908)	Antineoplastic (0.833)	Antifungal (0.714)
**138**	Antiprotozoal (Plasmodium) (0.970)	Apoptosis agonist (0.920)	Antifungal (Candida) (0.908)
	Antiparasitic (0.867)	Antineoplastic (0.850)	Antifungal (Cryptococcus) (0.844)
		DNA synthesis inhibitor (0.687)	Antifungal (0.812)
**139**	Antiprotozoal (Plasmodium) (0.972)	Apoptosis agonist (0.938)	Antifungal (Candida) (0.903)
	Antiparasitic (0.864)	DNA synthesis inhibitor (0.754)	Antifungal (0.833)
**140**	Antiprotozoal (Plasmodium) (0.977)	Antineoplastic (0.914)	Antifungal (Candida) (0.899)
		DNA synthesis inhibitor (0.733)	Antifungal (0.834)
**141**	Antiprotozoal (Plasmodium) (0.894)	Antineoplastic (0.943)	Anti-inflammatory (0.883)
**142**	Antiprotozoal (Plasmodium) (0.983)	Apoptosis agonist (0.955)	Antifungal (0.858)
	Antiparasitic (0.868)	Antineoplastic (0.841)	Antibacterial (0.633)
		DNA synthesis inhibitor (0.712)	
**143**	Antiprotozoal (Plasmodium) (0.988)	Apoptosis agonist (0.950)	Antifungal (0.860)
	Antiparasitic (0.859)	Antineoplastic (0.848)	Antibacterial (0.635)
		DNA synthesis inhibitor (0.718)	

* Only activities with Pa > 0.5 are shown.

**Table 10 molecules-26-00686-t010:** Biological activity of natural polycyclic peroxides derived from plants.

No	Antiprotozoal Activity, (Pa) *	Anticancer and Related Activities, (Pa) *	Additional Biological Activities, (Pa) *
**144**	Antiprotozoal (Plasmodium) (0.900)	Antineoplastic (0.873)	Respiratory analeptic (0.635)
		Apoptosis agonist (0.850)	Anti-psoriatic (0.630)
**145**	Antiprotozoal (Plasmodium) (0.893)	Antineoplastic (0.873)	Respiratory analeptic (0.642)
		Apoptosis agonist (0.852)	Anti-psoriatic (0.639)
**146**	Antiprotozoal (Plasmodium) (0.846)	Antineoplastic (0.857)	Antifungal (0.685)
**147**	Antiprotozoal (Plasmodium) (0.888)	Antineoplastic (0.824)	Choleretic (0.545)
		Chemopreventive (0.675)	Immunosuppressant (0.532)
		Apoptosis agonist (0.750)	
**148**	Antiprotozoal (Plasmodium) (0.734)	Antineoplastic (0.798)	
	Antiparasitic (0.544)		
**149**	Antiprotozoal (Plasmodium) (0.801)	Antineoplastic (0.871)	Prostate disorders treatment (0.627)
**150**	Antiprotozoal (Plasmodium) (0.915)	Antineoplastic (0.848)	Antibacterial (0.647)
**151**	Antiprotozoal (Plasmodium) (0.772)	Antineoplastic (0.726)	Antileukemic (0.602)
**152**	Antiprotozoal (Plasmodium) (0.829)	Antineoplastic (0.685)	Anti-eczematic (0.794)
**153**	Antiprotozoal (Plasmodium) (0.905)	Antineoplastic (0.805)	Carminative (0.505)
	Antiparasitic (0.727)		Antibacterial (0.504)
**154**	Antiprotozoal (Plasmodium) (0.854)	Antineoplastic (0.804)	Antimitotic (0.597)
			Antifungal (0.553)

* Only activities with Pa > 0.5 are shown.

**Table 11 molecules-26-00686-t011:** Biological activity of natural polycyclic peroxides derived from plants.

No	Antiprotozoal Activity, (Pa) *	Anticancer and Related Activities, (Pa) *	Additional Biological Activities, (Pa) *
**155**	Antiprotozoal (Plasmodium) (0.861)	Antineoplastic (0.788)	Antioxidant (0.551)
	Antiparasitic (0.503)	Apoptosis agonist (0.784)	Antibacterial (0.505)
		Chemopreventive (0.521)	
**156**	Antiprotozoal (Plasmodium) (0.862)	Antineoplastic (0.767)	Hypolipemic (0.581)
		Apoptosis agonist (0.744)	
**157**	Antiprotozoal (Plasmodium) (0.874)	Antineoplastic (0.761)	Antiviral (Arbovirus) (0.579)
		Apoptosis agonist (0.743)	
**158**	Antiprotozoal (Plasmodium) (0.875)	Antineoplastic (0.735)	
		Apoptosis agonist (0.675)	
**159**	Antiprotozoal (Plasmodium) (0.861)	Antineoplastic (0.788)	Antioxidant (0.551)
	Antiparasitic (0.503)	Apoptosis agonist (0.784)	Antibacterial (0.505)
		Chemopreventive (0.521)	
**160**	Antiprotozoal (Plasmodium) (0.875)	Antineoplastic (0.735)	
		Apoptosis agonist (0.675)	
**161**	Antiprotozoal (Plasmodium) (0.874)	Antineoplastic (0.761)	Antiviral (Arbovirus) (0.579)
		Apoptosis agonist (0.743)	
**162**	Antiprotozoal (Plasmodium) (0.811)	Apoptosis agonist (0.892)	Antioxidant (0.657)
		Antineoplastic (0.710)	
**163**	Antiparasitic (0.798)	Antineoplastic (0.958)	Antifungal (0.867)
	Antiprotozoal (Plasmodium) (0.731)	Apoptosis agonist (0.630)	Antibacterial (0.864)
	Antiprotozoal (Leishmania) (0.557)	Cytostatic (0.576)	Immunosuppressant (0.797)
**164**	Antiparasitic (0.779)	Antineoplastic (0.960)	Respiratory analeptic (0.879)
	Antiprotozoal (Plasmodium) (0.744)	Proliferative diseases treatment (0.740)	Immunosuppressant (0.754)
		Chemopreventive (0.666)	Angiogenesis inhibitor (0.569)
		Apoptosis agonist (0.627)	
**165**	Antiparasitic (0.771)	Antineoplastic (0.963)	Respiratory analeptic (0.879)
	Antiprotozoal (Plasmodium) (0.737)	Proliferative diseases treatment (0.742)	Immunosuppressant (0.754)
		Chemopreventive (0.656)	Angiogenesis inhibitor (0.569)
		Apoptosis agonist (0.629)	
**166**	Antiparasitic (0.798)	Antineoplastic (0.959)	Respiratory analeptic (0.936)
	Antiprotozoal (Plasmodium) (0.732)	Anticarcinogenic (0.732)	Immunosuppressant (0.773)
		Chemopreventive (0.731)	Angiogenesis inhibitor (0.620)
		Apoptosis agonist (0.622)	
**167**	Antiparasitic (0.813)	Antineoplastic (0.960)	Immunosuppressant (0.781)
	Antiprotozoal (Plasmodium) (0.728)	Chemopreventive (0.740)	Anti-inflammatory (0.765)
		Apoptosis agonist (0.631)	Analeptic (0.788)
**168**	Antiparasitic (0.843)	Antineoplastic (0.962)	Respiratory analeptic (0.964)
	Antiprotozoal (Plasmodium) (0.771)	Proliferative diseases treatment (0.834)	Neuroprotector (0.675)
		Apoptosis agonist (0.666)	
		Antimetastatic (0.517)	
**169**	Antiprotozoal (Plasmodium) (0.874)	Antineoplastic (0.761)	Antiviral (Arbovirus) (0.579)
		Apoptosis agonist (0.743)	
**170**	Antiprotozoal (Plasmodium) (0.875)	Antineoplastic (0.735)	
		Apoptosis agonist (0.675)	
**171**	Antiprotozoal (Plasmodium) (0.861)	Antineoplastic (0.788)	Antioxidant (0.551)
	Antiparasitic (0.503)	Apoptosis agonist (0.784)	Antibacterial (0.505)
		Chemopreventive (0.521)	

* Only activities with Pa > 0.5 are shown.

## Data Availability

Authors agree with the Publication Ethics Statement.

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
