# Peer review of "Antiprotozoal and Antitumor Activity of Natural Polycyclic Endoperoxides: Origin, Structures and Biological Activity"

_molecules, 2021, doi:10.3390/molecules26030686_

Round 1

Reviewer 1 Report

The manuscript presented more than 170 polycyclic endoperoxides isolated from various natural sources sources and their Antiprotozoal and Antitumor activities. The manu is well organized, only minor comments:

  • Use the past sentence in writing
  • there are free spaces in all tables which can be deleted
  • are you sure of the structure of compound rauianodoxy (133)? it has 3 oxygen atoms attached to each other, this is strange !
  • page 25: first paragrapgh "The PASS (Prediction of Activity Spectra for Substances) is the first software for in silico estimation of biological activity profiles [33,255], which development has been started more than thirty years ago [256].", you donnot need to defend PASS. In the same page, "The idea of using the PASS algorithm for testing natural drug-like compounds that are now used in clinical practice was born a long time ago. In this study, it was successfully implemented.", also delete it.
  • page 26, ligend of Fig 12, "Figure 12. The 3D graph shows predicted and calculated the pharmacological .." add of before the.
  • page 29, last paragraph of conclusion "Also compounds such as (19), (23) and (25) have been found that have anti-hyper-cholesterolemic action" replace "have been found that have" by "exhibited"
  • page 29, ref 1, 2008 is bold, make it similar to the others

Author Response

Reply to Reviewers

20 January 2021

Lethbridge, Canada

Dear Reviewers,

The authors have carefully read all the comments and recommendations of reviewers 1 and 2.

All comments have been corrected right in the text. The tables are in the required state.

The chemical structure of 133, taken from the original source, does contain 3 linked oxygen atoms. We have not found an error in this structure in the literature. In addition, other misprints and shortcomings have been corrected. To correct errors and improve the English language, we sent the article to the Department of English Philology, Lethbridge College, Canada. The quality of the English of our article is guaranteed.

Sincerely yours

Valery M Dembitsky

Lethbridge College

Lethbridge, Canada

Reviewer 2 Report

In this review, the authors presented more than 170 polycyclic endoperoxides isolated from various marine plants, fungi and invertebrates, carrying out a comparative analysis of the pharmacological potential of these natural products using the PASS (Prediction of Activity Spectra for Substances). This program expects these substances to have antiprotozoal and anticancer properties. All polycyclic endoperoxides presented in this article demonstrate antiprotozoal activity and some of them are widely used in clinical medicine. Furthermore, according to the PASS, some of the polycyclic endoperoxides analyzed, in addition to the antiprotozoal and anticancer properties, should have antifungal activity and anti-inflammatory activity. These predictions require experimental verification, but they certainly give a significant stimulus to the further study of polycyclic endoperoxides by medicinal chemists and pharmacologists and may arouse the interest of the pharmaceutical industry.

Author Response

(The authors gave the same response as above.)
